# Pianist Transformer: Towards Expressive Piano Performance Rendering via Scalable Self-Supervised Pre-Training

## Abstract

Existing methods for expressive music performance rendering rely on supervised learning over small labeled datasets, which limits scaling of both data volume and model size, despite the availability of vast unlabeled music, as in vision and language. To address this gap, we introduce Pianist Transformer, with four key contributions: 1) a unified Musical Instrument Digital Interface (MIDI) data representation for learning the shared principles of musical structure and expression without explicit annotation; 2) an efficient asymmetric architecture, enabling longer contexts and faster inference without sacrificing rendering quality; 3) a self-supervised pre-training pipeline with 10B tokens and 135M-parameter model, unlocking data and model scaling advantages for expressive performance rendering; 4) a state-of-the-art performance model, which achieves strong objective metrics and human-level subjective ratings. Overall, Pianist Transformer establishes a scalable path toward human-like performance synthesis in the music domain.

## 1 Introduction

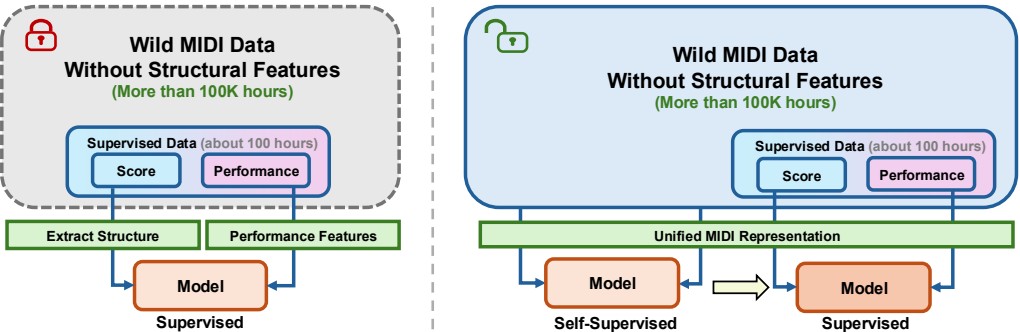

Figure 1: **The Paradigm Shift in Expressive Piano Performance Rendering. (Left) Previous Supervised Paradigm:** Existing systems operate under a strictly supervised pipeline that depends on scarce aligned datasets ($\approx 100$ hours) and cannot exploit the vast in-the-wild MIDI corpus ($> 100K$ hours). This reliance on explicit structural features fundamentally limits scalability. **(Right) Our Scalable Self-Supervised Paradigm:** Pianist Transformer shifts the paradigm by making large-scale self-supervised learning feasible for expressive piano performance rendering. Through the unified MIDI representation, the model can pre-train on over 100K hours of unaligned MIDI to acquire rich musical priors, and then generalize effectively through supervised fine-tuning.

Expressive performance rendering aims to automatically generate a human-like musical performance from a symbolic score. This task goes beyond mere pitch-and-rhythm accuracy to capture the subtle variations in timing, dynamics, articulation, and pedaling that shape musical expression. The core challenge lies in computationally modeling the intricate mapping from a score's underlying musical structures, such as its melody and harmony, to these expressive choices. For decades, research from probabilistic models (Teramura et al., 2008) to modern deep learning approaches using

RNNs (Jeong et al., 2019b) and Transformers (Borovik & Viro, 2023) has predominantly relied on a supervised paradigm. This paradigm, however, faces a persistent bottleneck: the aligned score-performance supervised datasets are typically labor-intensive and expensive to scale.

To maximize the utilization of the limited dataset, existing works often adopt asymmetric, specialized representations, injecting rich structural descriptors on the score side (e.g., measures, meter) (Jeong et al., 2019b; Maezawa et al., 2019; Borovik & Viro, 2023). This improves label efficiency because each labeled example delivers explicit structural cues rather than forcing the model to infer them. However, these descriptors require a notated score and a score–performance alignment. In contrast, performance MIDI is typically captured from digital-piano recordings or produced by AI transcription and thus consists of a stream of note events without explicit measures, meter, or a usable tempo map. As a result, such methods cannot compute their required structural features for the vast, unaligned corpora of performance-only MIDI, making them ill-suited for leveraging unsupervised data at scale (Figure 1, left). Renault et al. (2023) explores an adversarial, cycle-consistent architecture that disentangles score content from performance style, and a score-to-audio generator learns to render expressive piano audio from unaligned data against a realism discriminator. Nevertheless, adversarial training is challenging to scale due to complex training dynamics, and the resulting generation quality has so far been limited. A more stable and scalable paradigm is desirable to truly harness the potential of the vast, unaligned corpora of in-the-wild MIDI performances that remain largely untapped (Figure 1, right).

In this paper, we present Pianist Transformer, a model for expressive performance rendering trained with large-scale unlabeled MIDI corpus. Our main contributions are as follows:

**Unified Data Representation**: We introduce a single, fine-grained MIDI tokenization that encodes notated scores and expressive performances in the same discrete event vocabulary. By closing the representation gap between these modalities, this shared formulation makes unaligned, performance-only MIDI directly usable for pre-training, scaling to 10B MIDI tokens without explicit score-performance alignment, while preserving data diversity. In this unified space, the model can learn not only the "grammar" of music but also the statistical links between score-level structure and expressive controls (timing, dynamics, articulation, and pedaling).

**Efficient Architecture for Musical Modeling**: We design an asymmetric encoder-decoder architecture with note-level sequence compression that merges the fixed per-note event bundle into one token, reducing encoder self-attention cost by $64\times$. This concentrates compute in a single parallel pass, alleviates the decoding bottleneck, and yields longer context coverage and faster inference with strong rendering quality. Compared with a symmetric architecture, it delivers $2.1\times$ faster inference, meeting low-latency requirements for real-world use without sacrificing expressive quality.

**Scalable Training Pipeline**: We adopt a self-supervised pre-training scheme that furnishes the model with an initialization, internalizing common musical regularities and expressive patterns. Consequently, during downstream supervised fine-tuning the model starts from a stronger representation, converges faster and to a substantially lower loss than an otherwise identical model trained from scratch, and achieves stronger objective metrics. In particular, a scratch model tends to plateau at a higher loss and produces weaker expressive distributions, whereas the pre-trained model begins from a superior foundation and continues to improve throughout fine-tuning. To bridge model generation and practical music-production workflows, we introduce Expressive Tempo Mapping, a post-processing algorithm that converts model outputs into editable tempo maps. This produces an editable format suitable for real-world use while preserving expressive timing.

**Resulting performance model**: The overall training recipe yields a state-of-the-art expressive performance model. On objective metrics, it outperforms strong baselines. In a comprehensive listening study, its outputs are statistically indistinguishable from a human pianist and are more preferred.

## 2 RELATED WORK

**Piano Performance Rendering.** The goal of performance rendering is to synthesize an expressive, human-like performance from a symbolic score. The field has evolved from early rule-based (Sundberg et al., 1983) and statistical models (Teramura et al., 2008; Flossmann et al., 2013; Kim et al., 2013) to deep learning architectures based on RNNs (Cancino-Chacón & Grachten, 2016), variational autoencoders (Maezawa et al., 2019), graph neural networks (Jeong et al., 2019b), and

Transformers. For instance, recent work such as ScorePerformer (Borovik & Viro, 2023) has focused on fine-grained stylistic control, a goal complementary to our focus on scalable pre-training. Despite these architectural innovations, progress has been bottlenecked by the supervised learning paradigm; the small, costly aligned datasets it requires are insufficient for models to learn the complex mapping from musical structure to expressive nuance. This reliance limits model scalability and generalization. While recent work has explored adversarial training on unpaired data to bypass alignment (Renault et al., 2023), challenges with training stability and quality remain. This underscores the need for a robust paradigm that can effectively leverage vast, unaligned data, which we propose through large-scale self-supervised pre-training.

**Self-Supervised Learning in Music.** Self-supervised pre-training, a dominant paradigm in NLP (Devlin et al., 2019; Brown et al., 2020) and computer vision (Chen et al., 2020; He et al., 2022), has also been adapted for music. In the symbolic domain, early efforts like MusicBERT (Zeng et al., 2021) applied masked language modeling to MIDI for understanding tasks. This approach has recently been scaled up significantly: Bradshaw et al. (2025) pre-trained on large piano corpora for tasks like melody continuation, while foundation models like Moonbeam (Guo & Dixon, 2025) have been trained on billions of tokens for diverse conditional generation. Parallel efforts also exist for raw audio using contrastive or reconstruction objectives (Spijkervet & Burgoyne, 2021; Hawthorne et al., 2022). However, the application of self-supervised pre-training to the specific task of expressive performance rendering is largely unexplored. While existing self-supervised models excel at learning high-level musical semantics for tasks like generation or classification, performance rendering is a distinct, fine-grained challenge centered on modeling subtle expressive details. Whether the benefits of large-scale self-supervised pre-training can successfully transfer to this nuanced, performance-level domain is an open question that motivates our work.

## 3 THE PIANIST TRANSFORMER FRAMEWORK

Our goal is to develop a powerful piano performance rendering system that can leverage large-scale, unlabeled data through a self-supervised pre-training paradigm. This section details our approach, beginning with the core of our methodology: a unified data representation that enables large-scale pre-training. We then describe the Transformer-based architecture and the two-stage training strategy built upon this representation. Finally, we introduce a novel post-processing step that ensures the model's output is practical for musicians.

### 3.1 UNIFIED MIDI REPRESENTATION

A fundamental challenge in applying self-supervised learning to performance rendering is the disparity between structured score data and expressive performance data. Specifically, scores represent music with symbolic, metrical timing (e.g., quarter notes, eighth notes) and categorical dynamics (e.g., p, mf, f), while performances are captured as streams of events with absolute timing in milliseconds and continuous velocity values. To overcome this, we propose a unified, event-based token representation that treats both formats identically, enabling them to be mixed in a single, massive pre-training corpus.

We represent each musical note as a sequence of eight tokens. This sequence captures the note's Pitch, Velocity, Duration, and the Inter-Onset Interval (IOI) from the previous note, with timing information quantized from milliseconds. To model nuanced pedal control, we include four additional Pedal tokens, which represent the sustain pedal state at sampled points within the note's IOI window.

Crucially, this representation, avoiding reliance on high-level musical concepts like measures or beats, unlocks large-scale pre-training on unaligned MIDI and empowers the model to uncover musical principles, from melodic contours to harmonic progressions, through statistical regularities.

### 3.2 ARCHITECTURE FOR EFFICIENT LONG-SEQUENCE MODELING

We employ an Encoder-Decoder Transformer, but its standard $O(N^2)$ self-attention complexity presents a critical bottleneck for long musical sequences, which often exceed thousands of tokens. To enable efficient rendering, we introduce two synergistic architectural modifications: Encoder Sequence Compression and an Asymmetric Layer Allocation.

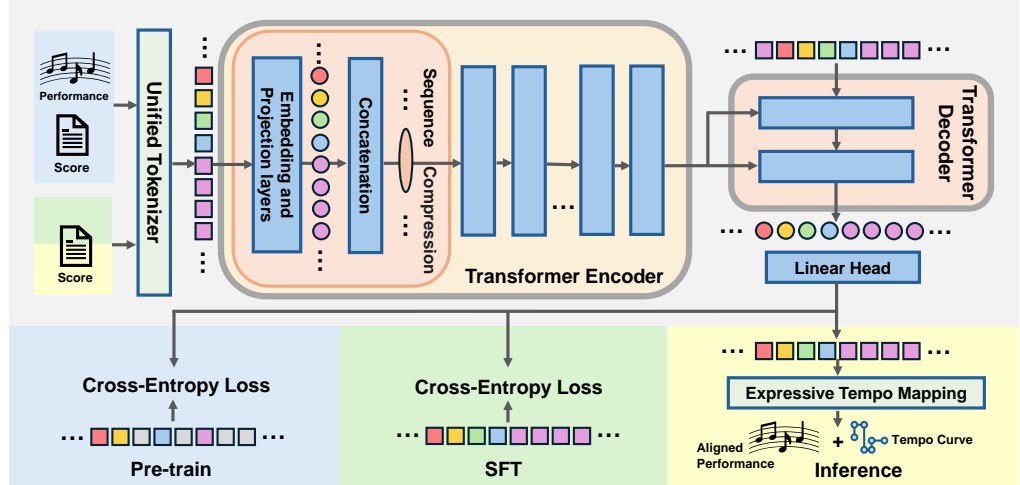

Figure 2: **The overall architecture and workflow of Pianist Transformer.** Our framework processes all MIDI data through a **Unified Tokenizer**, enabling a two-stage training process. The core model is an **asymmetric Transformer** with **Encoder Sequence Compression** for efficient processing of long musical scores. The workflow consists of three stages: **(1) Pre-train**: The model learns foundational musical context from a massive unlabeled corpus via a masked denoising objective, where it takes a masked token sequence as input and predicts the original sequence. **(2) SFT**: Supervised Fine-Tuning adapts the model to map musical context to expressive nuances using aligned score-performance pairs, where it takes the score tokens as input and predicts the corresponding performance tokens. **(3) Inference**: The model takes a score input and then generates a performance, which is then made editable for DAWs by our **Expressive Tempo Mapping** algorithm.

**Encoder Sequence Compression.** Leveraging the fixed 8-token structure of each note, we compress the encoder's input sequence. Instead of processing raw token embeddings, we first project and then aggregate the eight embeddings of a single note into one consolidated vector. This note-level aggregation reduces the sequence length by a factor of 8, which in turn leads to a 64-fold reduction in the self-attention computational cost from $O(N^2)$ to $O((N/8)^2)$. As a result, the encoder can efficiently process much longer sequence, capturing the global context essential for rendering.

**Asymmetric Encoder-Decoder Architecture.** We employ a deliberately asymmetric architecture with a deep 10-layer encoder with a lightweight 2-layer decoder (henceforth, 10-2) to maximize efficiency. This design, synergistic with our sequence compression, concentrates the majority of computation into a single, highly parallelizable encoding pass. This significantly accelerates training speed and reduces memory overhead for both training and inference. During generation, the shallow decoder, which is the primary bottleneck for autoregressive tasks, operates with minimal latency and memory footprint while being conditioned on the encoder's powerful representation. This architecture represents a conscious trade-off between computational efficiency and model performance, a balance which we quantitatively analyze in our ablation studies in Section 4.5.

### 3.3 TWO-STAGE TRAINING FOR EXPRESSIVE RENDERING

Our training paradigm directly addresses the core challenge of expressive rendering: modeling the complex dependency between a score's musical structure and the nuances of human performance. To achieve this, our training proceeds in two stages: first, learning to comprehend musical context, and second, learning to translate that context into an expressive performance.

#### 3.3.1 SELF-SUPERVISED PRE-TRAINING ON MUSICAL PRINCIPLES

The initial pre-training stage builds an understanding of the implicit context guiding human expression. We employ a self-supervised masked denoising objective on our massive, unlabeled MIDI

corpus. By learning to reconstruct the original music pieces from their corrupted context, the model is compelled to internalize the deep structural cues such as harmonic function and melodic direction that inform performance choices.

The objective is to minimize the negative log-likelihood of the original tokens at the masked positions:

$$\mathcal{L}_{\text{pre-train}} = - \sum_{i \in \mathbb{M}} \log p(x_i | \mathbb{X}_{corr}, \mathbb{X}_{<i})$$

where $\mathbb{M}$ is the set of indices of the masked tokens, and $p(x_i | \mathbb{X}_{corr}, \mathbb{X}_{<i})$ is the probability of predicting the original token $x_i$ given the corrupted input and the ground-truth prefix $\mathbb{X}_{<i}$.

### 3.3.2 SUPERVISED FINE-TUNING FOR EXPRESSIVE RENDERING

With a model that comprehends musical context, we then perform Supervised Fine-Tuning (SFT) to teach it how to translate this understanding into a performance. This stage learns the explicit mapping from the latent structural cues to the subtle, continuous parameters of human expression.

The SFT is framed as a sequence-to-sequence learning task on aligned score-performance pairs. The encoder processes the score's token sequence, while the decoder is trained to autoregressively generate the corresponding performance sequence by minimizing a standard cross-entropy loss. This fine-tuning stage grounds the model's expressive decisions, such as variations in timing and dynamics, in the deep musical understanding it acquired during pre-training.

### 3.4 POST-PROCESSING: EXPRESSIVE TEMPO MAPPING

A key challenge for practical application is that raw model outputs, with timings in absolute milliseconds, lack compatibility with standard music software. These performances do not align with the metrical grid of a Digital Audio Workstation (DAW), hindering editability. To bridge this gap between AI generation and modern music production workflows, we introduce a novel post-processing algorithm, Expressive Tempo Mapping.

This algorithm, detailed in Appendix C, intelligently translates the performance's expressive timing deviations into a dynamic tempo map. It then realigns all note and pedal events to a musical grid governed by this new tempo curve. The process preserves the sonic nuance of the generated performance while restoring the structural alignment essential for editing and integration. The final output is a MIDI file that is both musically expressive and fully editable in any standard DAW.

## 4 EXPERIMENTS

We conduct a comprehensive set of experiments to evaluate our proposed Pianist Transformer. Our evaluation is guided by three central questions. **First**, to what extent does large-scale self-supervised pre-training contribute to the final performance of a rendering model? **Second**, how does Pianist Transformer perform against existing methods when judged by both objective metrics and subjective human evaluation? **And third**, what architectural choices influence the model's effectiveness, and how robust is its performance across diverse musical contexts? The following sections are structured to address each of these questions in turn.

### 4.1 EXPERIMENTAL SETUP

We pre-train our model on a massive 10-billion-token corpus aggregated from several public MIDI datasets. For supervised fine-tuning and evaluation, we use the ASAP dataset (Foscarin et al., 2020) with a strict piece-wise split. Our Pianist Transformer is compared against strong baselines, including VirtuosoNet-HAN (Jeong et al., 2019a), VirtuosoNet-ISGN (Jeong et al., 2019b) and ScorePerformer (Borovik & Viro, 2023), as well as the unexpressive Score MIDI and ground-truth Human performances.

We evaluate all models using a suite of objective and subjective measures. Objectively, we assess distributional similarity to human performances using Jensen-Shannon (JS) Divergence and Intersection Area across four key expressive dimensions: Velocity, Duration, IOI and Pedal. Subjectively,

Table 1: **Objective evaluation results on the ASAP test set.** We compare our **Pianist Transformer** against baselines using JS Divergence and Intersection Area. For JS Div, lower is better (↓). For Intersection, higher is better (↑). The "Overall" columns report the average scores across the four expressive dimensions. Our model achieves the best performance on most metrics among the generative models, outperforming prior SOTA and demonstrating the profound impact of pre-training.

| Model | Velocity | | Duration | | IOI | | Pedal | | Overall (Avg.) | |
|---|---|---|---|---|---|---|---|---|---|---|
| | JS Div (↓) | Inter. (↑) | JS Div (↓) | Inter. (↑) | JS Div (↓) | Inter. (↑) | JS Div (↓) | Inter. (↑) | JS Div (↓) | Inter. (↑) |
| Human | 0.0427 | 0.9724 | 0.0438 | 0.9655 | 0.0535 | 0.9496 | 0.0244 | 0.9771 | 0.0411 | 0.9662 |
| Score | 0.7492 | 0.2255 | 0.6868 | 0.3152 | 0.7706 | 0.2106 | 0.4281 | 0.5467 | 0.6587 | 0.3245 |
| ScorePerformer | 0.4004 | 0.6256 | 0.3116 | 0.7126 | 0.4555 | 0.6071 | 0.6174 | 0.4164 | 0.4462 | 0.5904 |
| VirtuosoNet-ISGN | 0.2574 | 0.7981 | 0.2321 | 0.7903 | 0.5441 | 0.4928 | **0.0829** | **0.9410** | 0.2791 | 0.7556 |
| VirtuosoNet-Han | 0.2407 | 0.8132 | 0.3438 | 0.6744 | 0.4170 | 0.6374 | 0.1339 | 0.8507 | 0.2839 | 0.7439 |
| Pianist Transformer (w/o PT) | 0.5363 | 0.4826 | 0.5399 | 0.4886 | 0.2789 | 0.7360 | 0.2860 | 0.7054 | 0.4103 | 0.6032 |
| **Pianist Transformer (Ours)** | **0.1805** | **0.8517** | **0.1879** | **0.8303** | **0.1740** | **0.8292** | 0.1111 | 0.8893 | **0.1634** | **0.8501** |

we conduct a comprehensive listening study to evaluate human-likeness and overall preference. Comprehensive details regarding the datasets, baseline implementations, and evaluation protocols are provided in Appendix B and Appendix D.

## 4.2 Pre-training Substantially Improves Performance

To quantify the impact of large-scale self-supervised pre-training, we perform a controlled ablation comparing our full Pianist Transformer with an identical model trained from scratch (w/o PT). This setup isolates the effect of pre-training and reveals its crucial role in expressive performance modeling.

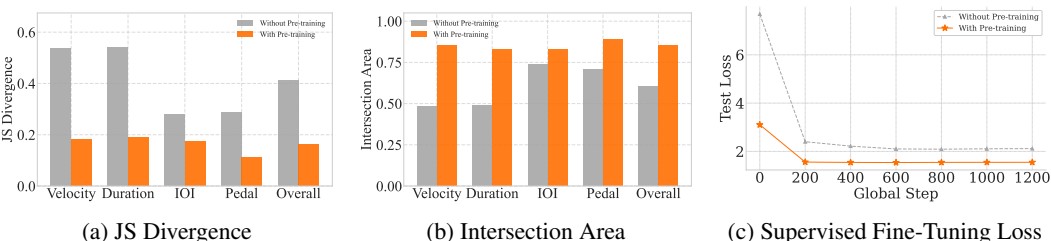

| (a) JS Divergence | (b) Intersection Area | (c) Supervised Fine-Tuning Loss |

Figure 3: **The profound impact of large-scale self-supervised pre-training.** We compare our **Pianist Transformer** against an identical model trained from scratch (w/o PT). (a, b) Pre-training leads to dramatic improvements in objective metrics that measure distributional similarity to human performances. (c) This is rooted in a much better learning foundation, as the pre-trained model converges faster and to a significantly lower loss during fine-tuning.

As shown in Figure 3, pre-training yields substantial improvements across both objective metrics and learning dynamics. Pre-training dramatically reduces JS Divergence and increases Intersection Area across all expressive dimensions (Figure 3a, 3b), indicating a significantly closer match to human performance distributions. The fine-tuning curves further reveal that the pre-trained model converges faster and reaches a much lower supervised loss, highlighting a superior initialization (Figure 3c).

The quantitative results in Table 1 further reinforce this gap. The overall Intersection Area improves from 0.6032 (scratch) to 0.8501 (pre-trained), representing a 40.9% relative gain. Pre-training also yields large reductions in JS Divergence for velocity (66.3%), duration (65.2%), IOI (37.6%), and pedal (61.2%). These consistent improvements across all expressive dimensions indicate that a model trained solely on limited supervised data struggles to capture the complex, high-variance distributions of human musicianship.

Together, these results validate that large-scale self-supervised pre-training is essential, as it provides the broad musical priors that limited supervised data alone cannot supply.

### 4.3 PIANIST TRANSFORMER ACHIEVES STATE-OF-THE-ART RESULTS

We now compare our full Pianist Transformer against prior state-of-the-art models. As shown in Table 1, our model demonstrates superior performance across the board.

Our model achieves the best scores among all generative models on 6 out of 8 metrics and on both overall average scores. Notably, Pianist Transformer significantly narrows the gap to the Human ground truth. For instance, its overall JS Divergence of 0.1634 represents a substantial improvement over the best baseline, VirtuosoNet-ISGN (0.2791). This indicates that the distributions of velocity, duration, and timing generated by our model are substantially more human-like than those from previous methods.

A closer look at the per-dimension results reveals our model's strengths in modeling musical time. The most significant gains are in Duration and IOI, which govern the rhythmic and temporal feel of the music. Our model's JS Divergence scores for these dimensions (0.1879 and 0.1740 respectively) are markedly lower than the best baseline scores. This suggests that large-scale pre-training endows the model with a sophisticated understanding of musical timing and phrasing, likely learned from the deep structural context in the data.

It is worth noting that VirtuosoNet-ISGN achieves a better score on the Pedal metrics, which may be attributed to its specialized architecture. However, our model still produces high-quality pedaling. Its JS Divergence of 0.1111 is competitive with the state-of-the-art (0.0829) and outperforms other baselines. This demonstrates that while not optimal in this specific dimension, our general-purpose pre-training approach yields strong all-around performance.

### 4.4 SUBJECTIVE EVALUATIONS REVEAL HUMAN-LEVEL PERFORMANCE

While objective metrics quantify statistical similarity, they often fail to capture the holistic qualities of a truly musical experience. We therefore conducted a comprehensive subjective listening study to perform a definitive, human-centric evaluation.

#### 4.4.1 STUDY DESIGN

We designed a rigorous subjective listening study to ensure the reliability and impartiality of our findings. We recruited 57 participants from diverse musical backgrounds and retained 39 high-quality responses after a stringent screening process based on attention checks and completion time. Participants rated and ranked five anonymized performance versions (our model, two baselines, Score, and Human) for six 15-second musical excerpts spanning Baroque to modern Pop styles. To mitigate bias, the presentation order of all performances was fully randomized for each participant. The comprehensive methodology, including participant demographics, stimuli selection, and our rigorous data validation protocols, is detailed in Appendix D.

#### 4.4.2 MAIN RESULTS: OVERALL PREFERENCE AND HUMAN-LIKENESS

The listening study results, summarized in Figure 4, reveal a clear and consistent preference for Pianist Transformer. The most direct measure of quality is listener preference, and our model was consistently ranked as the best among all generative systems.

As shown in Figure 4b, the first-place vote rate for Pianist Transformer (32.7%) was not only substantially higher than that of the baselines (7.7% and 14.7%) but was even marginally higher than the human pianist's (30.8%). This suggests that its renderings are not only realistic but also highly appealing to listeners.

This trend is corroborated by the average ranking results (Figure 4a). Pianist Transformer achieved the best overall average rank (2.29), outperforming the human performance (2.50). To rigorously assess these differences, we conducted a series of two-sided paired t-tests. The results confirm that our model is rated significantly better than VirtuosoNet-ISGN ($p < 0.001$), VirtuosoNet-Han ($p < 0.001$), and the Score baseline ($p < 0.001$). While the observed advantage of our model over the Human performance was not found to be statistically significant ($p = 0.21$), this result provides strong evidence that our model has reached a level of quality that is not only on par with but also highly competitive against human artists.

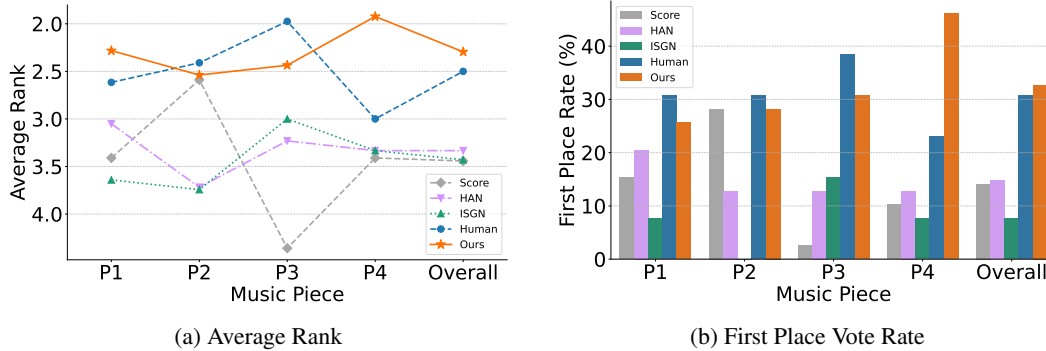

(a) Average Rank

(b) First Place Vote Rate

Figure 4: **Subjective Preference Ranking Results.** The evaluation includes pieces by Haydn (P1), Beethoven (P2), Chopin (P3), and Bach (P4). (a) The average rank of our **Pianist Transformer** is statistically indistinguishable from the Human performance and significantly better than all baselines. (b) Our model achieves a slightly higher first-place vote rate than the human pianist, demonstrating strong listener appeal.

### 4.4.3 Multi-dimensional Quality and Stylistic Robustness

To understand the reasons behind this strong listener preference, we analyzed the multi-dimensional ratings and the model's performance across different musical styles within our test pieces.

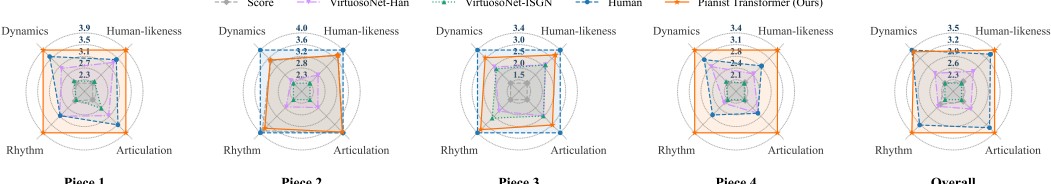

Figure 5: **Multi-dimensional Subjective Ratings (Normalized).** A radar chart visualizing the average scores on a 5-point scale for four expressive dimensions. **Pianist Transformer** exhibits a profile that closely mirrors the **Human** performance , indicating a well-balanced and high-quality rendering across all aspects. The area covered by ours is substantially larger than all of other baselines.

As visualized in the radar chart (Figure 5), the expressive profile of our Pianist Transformer closely mirrors that of the Human performance, indicating a well-balanced, high-quality rendering across all rated aspects. Quantitatively, our model's average scores were rated higher than the human pianist's not only in Rhythm & Timing (3.44 vs. 3.21) and Articulation (3.38 vs. 3.24), but also in the most critical global metric, Human-likeness (3.43 vs. 3.29). This remarkable result suggests our model generates performances that are perceived as human, perhaps even as idealized versions, free of the minor imperfections or idiosyncratic choices present in any single human recording.

Furthermore, the benefits of pre-training are evident in the model's stylistic robustness across historical periods, as shown in Figure 6. While baseline models exhibit a strong style dependency, with their performance degrading significantly for Baroque and Classical pieces, Pianist Transformer maintains a consistently high level of human-likeness across all styles, close to the human level. We attribute this robustness to the diverse musical knowledge acquired during large-scale pre-training, which prevents overfitting to the specific stylistic biases of the fine-tuning dataset. A case study on out-of-domain generalization to pop music is provided in Appendix D.3.

### 4.5 Analysis of Scaling Effects and Architecture

In our final analysis, we conduct a preliminary exploration of scaling effects to understand the relationship between performance, scale, and our architectural choices. The results, presented in Figure 7, validate that our framework is scalable while also revealing key bottlenecks that inform our design trade-offs.

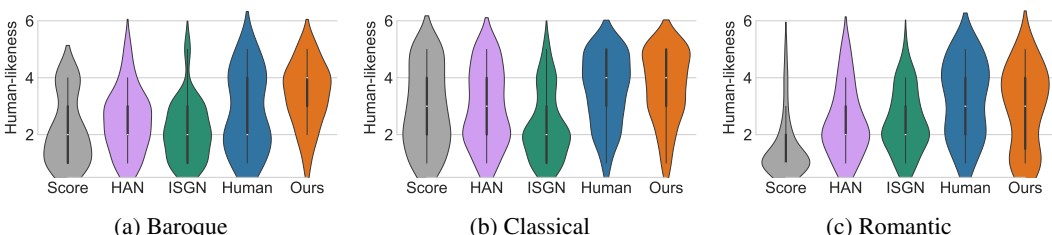

(a) Baroque      (b) Classical      (c) Romantic

Figure 6: **Analysis of Human-likeness Scores Across Musical Styles.** Violin plots show the distribution of Human-likeness ratings for each model, grouped by historical period. (a, b) For Baroque and Classical music, the performance of baseline models degrades significantly, sometimes falling below the Score baseline. (c) While baselines perform better on Romantic music, **Pianist Transformer** maintains a consistently high level of performance across all styles.

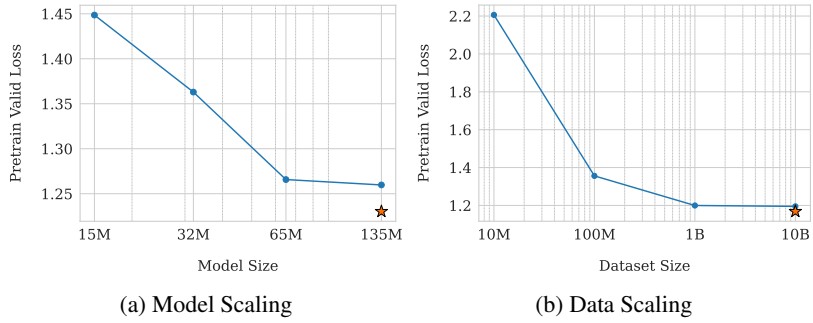

(a) Model Scaling      (b) Data Scaling

Figure 7: **A Preliminary Exploration of Scaling Effects.** Pre-training validation loss as a function of model size and data volume. (a) Increasing parameters in our asymmetric 10-2 architecture consistently improves performance, though saturation is observed at 135M. The star marks the lower loss achieved by a symmetric 6-6 variant, highlighting the decoder as a bottleneck. (b) Increasing data yields substantial gains up to 1B tokens, after which performance plateaus, suggesting the 135M model's capacity becomes a limiting factor.

**Model Scaling and Decoder Bottleneck.** We first analyze the effect of model size. As shown in Figure 7a, increasing model parameters from 15M to 65M yields substantial performance gains. However, the curve flattens significantly from 65M to 135M, indicating performance saturation. We hypothesized that this bottleneck stems from our lightweight 2-layer decoder. To test this, we trained a symmetric 6-layer encoder, 6-layer decoder (6-6) variant with a more powerful decoder (marked by a star). It achieved a notably lower loss of 1.230 compared to our 135M model's 1.260, confirming that the shallow decoder is indeed the primary bottleneck for model capacity scaling.

**Data Scaling and Model Capacity Bottleneck.** Next, we examine the impact of data scale. Figure 7b shows a dramatic drop in loss when scaling data up to 1B tokens. However, the performance again saturates when increasing the data tenfold to 10B (1.199 vs. 1.195). To determine if this was also a decoder issue, we leveraged our more powerful 6-6 model. Even with this stronger architecture, the loss on 10B data only marginally decreased to 1.168. This suggests that for a massive 10B-token dataset, the overall model capacity (around 135M parameters) itself becomes the primary bottleneck, regardless of the encoder-decoder layer allocation.

**Architectural Trade-off.** While the symmetric 6-6 model achieved a lower pre-training loss, confirming that the shallow decoder is a capacity bottleneck, our detailed evaluation in Appendix G reveals that this advantage did not translate to superior downstream performance. This suggests that fully capitalizing on the deeper decoder's potential may require more extensive training or data scaling than currently employed. Consequently, we prioritized the asymmetric 10-2 architecture. As demonstrated in the efficiency analysis in Table 10, the 10-2 model delivers comparable render-

ing quality while being approximately 2.1x faster during CPU inference, representing a far more practical trade-off for real-world applications.

## 5 CONCLUSION

In this work, we introduced Pianist Transformer, establishing a new state-of-the-art in expressive piano performance rendering through large-scale self-supervised pre-training. By learning from a 10-billion-token MIDI corpus with a unified representation, our model overcomes the data scarcity that has hindered prior methods from learning the complex mapping between musical structure and expression. Our experiments provide compelling evidence for this paradigm shift: Pianist Transformer not only excels on objective metrics but achieves a quality level statistically indistinguishable from human artists in subjective evaluations, where its renderings were sometimes preferred. Furthermore, our model demonstrates robust performance across diverse musical styles, a direct benefit of its pre-trained foundation. Ultimately, Pianist Transformer demonstrates that scaling self-supervised learning is a promising path toward generating music with genuine human-level artistry, establishing an effective and scalable paradigm for future research in computational music performance.

## ETHICS STATEMENT

The pre-training and fine-tuning of our model were conducted exclusively using publicly available datasets, as detailed in Appendix B. Our research did not involve the collection of new private data. For our subjective listening study, we recruited human participants. All participants were presented with an informed consent form prior to the study, which outlined the purpose of the research, the nature of the task, and how their data would be used. To protect participant privacy, all experimental responses were fully anonymized prior to analysis and were stored separately from any personal information required for compensation. We compensated each participant for their time and effort with a payment that exceeds the local minimum wage standard. We foresee no direct negative societal impacts resulting from this work, which is intended to advance research in computational music and creativity.

## REPRODUCIBILITY STATEMENT

We are fully committed to the reproducibility of our research. Upon publication, we will release our complete source code and final model weights. To facilitate comprehensive verification during the review process, the supplementary materials include the source code, a representative subset of audio renderings for the test set, as well as the audio samples used in our subjective listening study. Furthermore, the complete set of generated audio for the entire test set and all listening study materials are available for review at the following anonymous URL: `https://anonymous.4open.science/r/JSKJDHKIOWBBCGFBDKS/`. All essential details regarding our methodology, including data sources, model architecture, and hyperparameters, are thoroughly documented in the appendices of this paper.

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

TABLE OF CONTENTS FOR APPENDIX

## A  LLMs Usage Statement

We used Large Language Models (LLMs) to assist with the writing of this paper. Their primary role was to improve grammar, phrasing, and clarity. LLMs were also used to help survey and summarize related work. In accordance with ICLR policy, the authors have reviewed and take full responsibility for all content, including its scientific accuracy and any text influenced by the LLMs.

## B  Experimental Setup & Implementation Details

### B.1  Dataset Details

The pre-training corpus is constructed from the following sources. We applied specific preprocessing steps to ensure data quality and diversity.

- **Aria-MIDI** (Bradshaw & Colton, 2025): A large collection of over 1.1 million MIDI files transcribed from solo piano recordings. To ensure high fidelity, we only included segments with a transcription quality score above 0.95. Due to the coarse quantization in the original transcriptions, we applied random augmentations to the Velocity, Duration, and IOI values of these files to better simulate performance nuances.
- **GiantMIDI-Piano** (Kong et al., 2022): A dataset of over 10,000 unique classical piano works transcribed from live human performances using a high-resolution system. These files retain fine-grained expressive details, including velocity, timing, and pedal events.
- **PDMX** (Long et al., 2025): A diverse dataset of over 250,000 musical scores, originally in MusicXML format. We used their MIDI conversions to provide our model with clean, score-based MIDI data. To filter out overly simplistic or empty files, we only included MIDI files larger than 7 KB.
- **POP909** (Wang et al., 2020): A dataset of 909 popular songs. We extracted the piano accompaniment tracks to include non-classical and accompaniment-style patterns.
- **Pianist8** (Chou et al., 2021): A collection of 411 pieces from 8 distinct artists, consisting of audio recordings paired with machine-transcribed MIDI files.

For SFT and evaluation, we use the **ASAP dataset** (Foscarin et al., 2020), a collection of aligned score-performance pairs of classical piano music.

Before training, we first normalize all MIDI files so they can be tokenized under a consistent format. For multi-track scores, we merge all tracks into a single time-ordered event stream and remove duplicate notes created during merging. We also convert every file to a fixed tempo of 120 BPM and rescale all onset times and durations. After this processing, all MIDI files share the same temporal scale and event structure, allowing reliable and uniform tokenization across the entire corpus.

To ensure precise note-level correspondence between scores and performances, we refined the provided alignments. We first employed an HMM-based note alignment tool (Nakamura et al., 2017) to establish a direct mapping for each note. For localized mismatches where a few notes could not be paired, we applied an interpolation algorithm to infer the correct alignment based on the surrounding context. Finally, segments with large, contiguous blocks of unaligned notes were filtered out and excluded from our training and evaluation sets to maintain high data quality. We create a strict piece-wise split by randomly holding out 10% of the pieces for our test set. The remaining 90% are used for fine-tuning.

### B.2  Model Architecture and Tokenizer

#### B.2.1  Model Architecture

Our Pianist Transformer employs an asymmetric encoder-decoder architecture based on the T5-Gemma framework (Zhang et al., 2025). The encoder is designed to be substantially deeper than the decoder, with 10 layers, to efficiently process long input sequences and build a rich contextual representation. The decoder, with only 2 layers, is lightweight to ensure fast and efficient autoregressive generation during inference. This design strikes a balance between expressive power and practical utility. Key hyperparameters for our 135M model are detailed in Table 2.

Table 2: Key hyperparameters for the Pianist Transformer.

| Parameter | Value |
|---|---|
| Model Architecture | T5-Gemma |
| Total Parameters | $\approx$ 135M |
| Hidden Size | 768 |
| Intermediate Size (FFN) | 3072 |
| Number of Encoder Layers | 10 |
| Number of Decoder Layers | 2 |
| Attention Head Dimension | 128 |
| Total Vocabulary Size | 5389 |

### B.2.2 UNIFIED MIDI REPRESENTATION

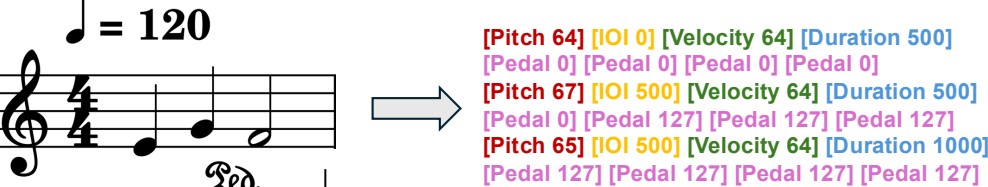

Figure 8: An Example of the Unified MIDI Representation

Central to our approach is a unified, event-based token representation that treats both score and performance MIDI identically. Each musical note is represented as a fixed-length sequence of eight tokens, capturing its core attributes and nuanced pedal information. The sequence order is: `[Pitch, IOI, Velocity, Duration, Pedal1, Pedal2, Pedal3, Pedal4]`.

The vocabulary is structured as follows:

- **Pitch**: MIDI pitch values are mapped directly to 128 tokens (range 0 to 127).
- **Velocity**: MIDI velocity values are mapped to 128 tokens (range 0 to 127).
- **Timing (IOI & Duration)**: The Inter-Onset Interval (IOI) and Duration are quantized at a 1ms resolution and share a common vocabulary of 5000 tokens. The Duration can utilize the full range (0 to 4999), while the IOI is restricted to a slightly smaller range (0 to 4990) to avoid a known artifact in transcribed MIDI where durations frequently saturate at the maximum value.
- **Pedal**: Four pedal tokens represent the sustain pedal state sampled at four equidistant points within the interval leading to the next note. Each pedal value is mapped to one of 128 tokens (range 0 to 127). While this representation supports continuous (half-pedal) values, our pre-training data predominantly contained binary pedal events (0 or 127), effectively training the model to generate on/off pedal control.

Special tokens including `[PAD]`, `[MASK]`, `[BOS]`, `[EOS]`, and a special `[PLAY]` unused, resulting in a total vocabulary size of 5389. Figure 8 provides a concrete example of how a short musical phrase, under 120 BPM, is mapped into our 8-token event representation capturing pitch, timing, velocity, and pedal states.

### B.2.3 COMPARISON WITH PRIOR MIDI TOKENIZATION SCHEMES

Table 3 provides a comparison between our MIDI representation and several prior tokenization schemes. The design of our representation is guided by the specific requirements of the expressive performance rendering task, particularly the need to leverage large-scale MIDI data without structural features.

The core advantages of our representation are as follows:

Table 3: Comparison of MIDI tokenization schemes. Our representation is tailored for scalable, self-supervised expressive performance rendering.

| Representation | Temporal Representation | Note-centric | Encodes Pedal |
|---|---|---|---|
| REMI (Huang & Yang, 2020) | Bar & Pos | No | No |
| MIDI-Like (Oore et al., 2020) | Time shift | No | No |
| CPWord (Hsiao et al., 2021) | Bar & Pos | No | No |
| Octuple (Zeng et al., 2021) | Bar & Pos | Yes | No |
| **Ours** | **Time shift** | **Yes** | **Yes** |

- **Unlocks Self-Supervised Pre-training.** By using time shift for temporal representation, our method does not require structural features like bars or beats. This is a crucial advantage because it allows us to pre-train on massive datasets of performance MIDI, which lack this structure and thus cannot be used by other methods.

- **Designed for Performance Rendering.** Our note-centric approach treats each note and its properties as a single unit. This design is a natural fit for the rendering task, as it simplifies the process of matching an input score to an output performance and makes our model highly efficient.

- **Captures Essential Piano Acoustics.** Our representation explicitly encodes the sustain pedal, a crucial factor in producing the rich resonance characteristic of real piano performances. Incorporating pedal information enables the model to generate more expressive and realistic renderings.

### B.3 TRAINING PROCEDURE

#### B.3.1 SELF-SUPERVISED PRE-TRAINING

The pre-training phase is designed to build a foundational understanding of musical structure and expression from our large-scale, unlabeled MIDI corpus. We employ a masked denoising objective, similar to T5, where the model learns to reconstruct corrupted segments of the input token sequences. In this stage, following recent mature practices in NLP for masked denoising objectives (Warner et al., 2025) we adopt a masking ratio of 0.3 with tokens randomly masked. The model was trained for 40,000 steps using the AdamW optimizer (Loshchilov & Hutter, 2019). Key hyperparameters for this stage are detailed in Table 4.

#### B.3.2 SUPERVISED FINE-TUNING

The fine-tuning process ran for 2 epochs. We adopted a slightly higher learning rate than in pre-training, which empirically led to faster convergence and a lower final loss. The learning rate followed a cosine decay schedule without a warmup phase. The global batch size was set to 32. All other settings remained consistent with the pre-training stage. A side-by-side comparison of pre-training and SFT hyperparameters is provided in Table 4.

Table 4: Comparison of hyperparameters for Pre-training and SFT stages.

| Parameter | Pre-training | SFT |
|---|---|---|
| Optimizer | AdamW | |
| Learning Rate Schedule | Cosine decay | |
| Peak Learning Rate | 3e-4 | 5e-4 |
| Warmup Steps | 2,500 | 0 |
| Training Duration | 40,000 steps | 2 epochs |
| Global Batch Size | 64 | 32 |
| Maximum Sequence Length | 4096 | |
| Precision | bfloat16 | |
| Hardware | 4x NVIDIA A800 GPUs | |

### B.4 CALCULATION OF OBJECTIVE METRICS

To measure how closely the generated performances resemble human playing, we compare the global token distributions of the model outputs with those of the human performances across the entire test set. For Velocity, Duration, and IOI, we aggregate the tokens of each type from all generated pieces and compute their distributions, which we then compare with the corresponding human distributions using JS Divergence and Intersection Area. For Pedal, where the corpus mainly contains binary values, we binarize both model outputs and human data and evaluate the distribution of the 16 possible joint configurations formed by the four pedal tokens of each note.

To obtain a human baseline, for every piece, we treat one human performance as the candidate and use the remaining human performances as the reference set, applying the same distributional comparison. This provides a measure of the natural stylistic variation among human performers.

### B.5 BASELINE IMPLEMENTATION

For VirtuosoNet-HAN and VirtuosoNet-ISGN, we used the official implementations and pre-trained weights, followed their recommended inference procedures, and selected the composer-style configurations that best matched the pieces in our test set.

ScorePerformer was evaluated under the same score-only setting. Since it is designed to operate with fine-grained style vectors derived from reference performances, which are not available in our setup, we adopted the unconditional generation mode recommended in the original paper, where style vectors are sampled from the prior distribution.

All generated MIDI files from all models were rendered to audio using the same high-quality piano soundfont to ensure a fair subjective listening study.

## C EXPRESSIVE TEMPO MAPPING ALGORITHM

To make our model's output compatible with standard music production software, we introduce the Expressive Tempo Mapping algorithm. This process converts the generated performance, which has timing in absolute milliseconds, into a standard MIDI file where expressive timing is encoded as a dynamic tempo map. This makes the performance fully editable within any DAW. The procedure is outlined in Algorithm 1.

---

**Algorithm 1** Expressive Tempo Mapping

1: **Input:** Score MIDI $M_{score}$, Performance MIDI $M_{perf}$
2: **Output:** DAW-friendly expressive MIDI $M_{DAW}$
3: Extract notes $N_{score}, N_{perf}$ and pedal events $CC_{perf}$ from input files.
4: Estimate a dynamic tempo curve $T_{changes}$ based on timing deviations.
5: Initialize empty lists for aligned events: $N_{aligned}, CC_{aligned}$.
6: **for** each corresponding note pair $(n_{score}, n_{perf})$ **do**
7:     Create a new note $n_{new}$ where:
8:         - pitch is from pitch of $n_{score}$
9:         - velocity is from velocity of $n_{perf}$
10:         - onset in ticks is converted from $n_{perf}$'s onset in milliseconds using $T_{changes}$.
11:         - duration in ticks is converted from $n_{perf}$'s duration in milliseconds using $T_{changes}$.
12:     Append $n_{new}$ to $N_{aligned}$.
13: **end for**
14: **for** each control event $cc$ in $CC_{perf}$ **do**
15:     Convert $cc$'s timestamp from milliseconds to ticks using $T_{changes}$ to get $t_{new}$.
16:     Create a new control event $cc_{new}$ with value from $cc$ and time from $t_{new}$.
17:     Append $cc_{new}$ to $CC_{aligned}$.
18: **end for**
19: Assemble $M_{DAW}$ by combining $T_{changes}, N_{aligned}$, and $CC_{aligned}$.
20: **return** $M_{DAW}$

---

The algorithm executes in three main stages:

1. **Tempo Estimation (Line 4):** First, we compare the timing of note onsets between the score MIDI ($M_{\text{score}}$) and the generated performance MIDI ($M_{\text{perf}}$). The differences in timing are used to calculate a local tempo (BPM) for each segment of the piece. This sequence of tempo changes forms a dynamic tempo curve, $T_{\text{changes}}$, which captures all the expressive timing (rubato) of the performance.

2. **Event Remapping (Lines 6-18):** Next, we create a new set of notes and pedal events. Each new note uses the pitch from the original score and the velocity from the generated performance. The crucial step is converting the onset time and duration of every note and pedal event from absolute milliseconds into musical ticks. This conversion is done using the tempo curve $T_{\text{changes}}$ estimated in the previous step. This aligns all events to a musical grid while preserving their expressive timing.

3. **Final Assembly (Line 19):** Finally, the newly created tempo curve ($T_{\text{changes}}$), the remapped notes ($N_{\text{aligned}}$), and the remapped pedal events ($CC_{\text{aligned}}$) are combined into a single, standard MIDI file ($M_{\text{DAW}}$). The resulting file sounds identical to the original performance but is now fully editable in a DAW, with all timing nuances represented in the tempo track.

# D  SUBJECTIVE LISTENING STUDY DETAILS

To conduct a definitive, human-centric evaluation of our model's performance, we designed and carried out a comprehensive subjective listening study. This appendix provides a detailed account of the study's design, participants, materials, and procedures.

## D.1  PARTICIPANT DEMOGRAPHICS

Our subjective listening study's validity rests on the quality and diversity of its participant pool. We initially recruited 57 individuals; after a rigorous screening for attentiveness and completion quality, 39 responses were retained for the final analysis. This section details the demographic composition of this group, providing evidence for its suitability for the nuanced task of evaluating musical expression.

The detailed distributions of participants' musical experience and listening habits are visualized in Figure 9. Several key characteristics of the group bolster the credibility of our findings:

**Balanced Expertise Spectrum (Figure 9a).** The participants' formal music training is not skewed towards one extreme. The pool includes a substantial proportion of listeners with no formal training (28.2%), ensuring that our model's appeal is not limited to musically educated ears. Concurrently, the presence of highly experienced individuals (15.4% with $> 10$ years of training) guarantees that subtle expressive details are also being critically evaluated. This heterogeneity mitigates potential bias and strengthens the generalizability of our preference results.

**Representative Listening Habits (Figure 9b).** The distribution of classical piano listening frequency reflects a general audience rather than a niche group of connoisseurs. The largest segment listens "Monthly" (46.2%), suggesting that the superior performance of Pianist Transformer is perceptible and appreciated even by those who are not deeply immersed in the genre daily.

**Competent and Calibrated Self-Assessment (Figure 9c).** The self-assessed ability to discern music quality is centered around "Moderate" (46.2%), with a healthy portion rating themselves as "High" (20.5%). This distribution suggests a group that is confident in their judgments without being overconfident, indicating that the participants were well-suited for the evaluation task.

In summary, the participant pool is intentionally diverse, comprising a mix of novices, enthusiasts, and experts. This composition ensures that our findings are robust, reliable, and reflective of a broad range of listener perceptions.

## D.2  MUSICAL EXCERPTS FOR EVALUATION

The listening study was based on six musical excerpts, each approximately 15 seconds long. To ensure an unbiased comparison, all excerpts were systematically taken from the beginning of each

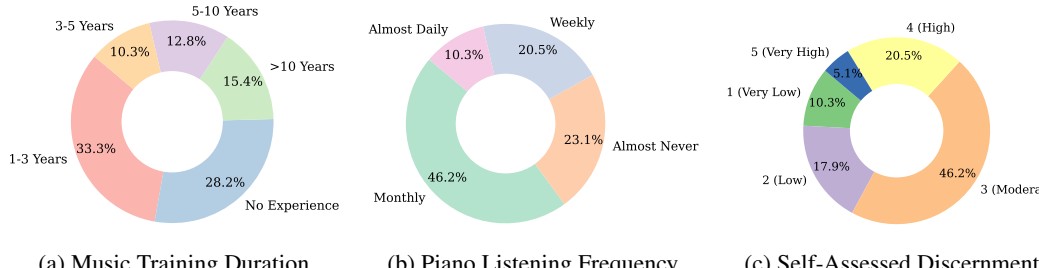

(a) Music Training Duration     (b) Piano Listening Frequency     (c) Self-Assessed Discernment

Figure 9: **Demographic distribution of the 39 participants in the listening study.** The plots show (a) the duration of formal music training, (b) the frequency of listening to classical piano music, and (c) self-assessed ability to discern piano music quality on a 1-5 scale. This diverse composition validates the generalizability of our study's findings.

piece. The selection was also deliberately curated for stylistic breadth, featuring works from the Baroque, Classical, and Romantic periods, as well as modern pop style. This diversity provides a rigorous testbed for evaluating the models' generalization abilities across varied musical contexts. The specific pieces are detailed in Table 5.

Table 5: Musical excerpts selected for the subjective listening study, highlighting their stylistic diversity.

| Composer | Work | Period / Style |
|---|---|---|
| J. S. Bach | Prelude and Fugue in G minor, BWV 885, Prelude | Baroque |
| J. Haydn | Keyboard Sonata No. 58 in C major, Hob.XVI:48:II | Classical |
| L. v. Beethoven | Piano Sonata No. 4 in E-flat major, Op. 7:I | Classical |
| F. Chopin | Étude in D-flat major, Op. 25, No. 8 | Romantic |
| F. Liszt | Étude d'exécution transcendante No. 1, Preludio, S. 139 | Romantic |
| Joe Hisaishi | Merry-Go-Round of Life | Modern Pop |

### D.3 Case Study: Generalization to Out-of-Domain Popular Music

To rigorously probe the generalization capabilities of our model, we included a musical excerpt from a modern popular song. This piece is stylistically distinct from the primarily classical ASAP dataset used for fine-tuning, thereby serving as a challenging out-of-domain test. The goal was to assess whether the robust musical understanding gained during pre-training would translate effectively to genres beyond the immediate scope of the fine-tuning data.

The results of this case study, summarized in Figure 10, reveal a nuanced dynamic. We observe that VirtuosoNet-ISGN delivers a highly competitive performance on this slow, lyrical piece. Its multi-dimensional ratings (Figure 10a) and average rank (Figure 10b) are nearly on par with our Pianist Transformer, suggesting that the expressive patterns it learned are well-suited for this particular style of song-like playing.

However, a crucial distinction emerges from the first-place vote rate (Figure 10c). Despite the close average scores, listeners chose Pianist Transformer as the single best performance by a dominant margin. This finding highlights a key advantage of our approach. While other systems may produce competent or even good performances on stylistically favorable pieces, Pianist Transformer is significantly more likely to generate a truly exceptional rendering that listeners perceive as the definitive best. We attribute this superior appeal to the fine-grained nuances and avoidance of subtle AI artifacts learned during large-scale pre-training, which ultimately translates to a more compelling and preferred musical experience.

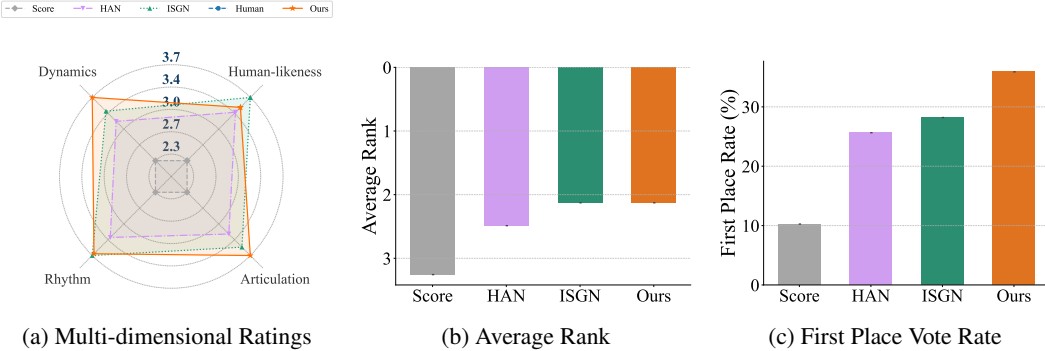

(a) Multi-dimensional Ratings      (b) Average Rank      (c) First Place Vote Rate

Figure 10: **Case Study on an Out-of-Domain Popular Music Excerpt.** Subjective evaluation results for a slow, lyrical pop piece. (a, b) VirtuosoNet-ISGN performs competitively in average ratings and rankings. (c) However, our **Pianist Transformer** secures a dominant share of first-place votes, indicating superior overall appeal and quality.

### D.4 RELIABILITY AND CONSISTENCY ANALYSIS

To ensure the robustness and impartiality of our subjective evaluation, we embedded an internal consistency check within the study. For one musical excerpt (Liszt's work), two identical audio clips from the same model's performance were presented to each participant as if they were distinct versions. The analysis of ratings for these duplicates, shown in Table 6, provides crucial insights into the study's validity.

First, the analysis confirms the experiment's impartiality. The Mean Error (Bias) between the ratings for the duplicate clips is negligible, and a paired t-test showed these differences to be statistically insignificant (all $p > 0.6$). This result demonstrates that our experimental design successfully mitigated systematic biases, such as those arising from presentation order or listener fatigue. Furthermore, the Pearson correlation ($r$) between the paired ratings is positive but modest. This is an expected outcome, reflecting the inherent variability and noise in the subjective human perception of music. The presence of this natural perceptual uncertainty makes our main findings, the clear and statistically significant preference for Pianist Transformer, even more compelling. It indicates that the perceived quality difference between our model and its counterparts was strong and consistent enough to overcome this noise, thereby solidifying the significance and reliability of our conclusions.

Table 6: Intra-rater reliability analysis on duplicate audio stimuli. We report the Mean Error (Bias) and Pearson Correlation ($r$) between ratings for two identical audio clips presented to the same user. The low, statistically non-significant bias confirms the experiment's impartiality.

| Metric | Dynamics | Rhythm | Articulation | Human-likeness |
|---|---|---|---|---|
| Mean Error (Bias) | 0.026 | -0.103 | 0.000 | 0.051 |
| Pearson Corr. ($r$) | 0.155 | 0.377 | 0.351 | 0.133 |

### E ABLATION STUDY OF THE MASKING RATIO IN PRE-TRAINING

To analyze the effect of the masking ratio on downstream performance, we conducted an ablation study by pre-training models with three different ratios: 15%, 30% and 45% and then fine-tuning them on our rendering task. The results are presented in Table 7.

The results indicate that our main setting of 30% outperforms the 15% ratio, while the 45% ratio yields slightly better overall performance. This suggests that the optimal masking strategy for symbolic music may differ from common practices in NLP, an interesting direction for future work.

However, all three pre-trained models significantly outperform the supervised-only baselines, showing that the gains from self-supervised pre-training do not depend on any specific masking setting.

Table 7: Ablation study on the pre-training masking ratio. While the 45% ratio achieves the best overall scores, all pre-trained variants significantly outperform supervised-only baselines.

| Mask Ratio | Velocity | | Duration | | IOI | | Pedal | | Overall (Avg.) | |
|---|---|---|---|---|---|---|---|---|---|---|
| | JS Div ($\downarrow$) | Inter. ($\uparrow$) | JS Div ($\downarrow$) | Inter. ($\uparrow$) | JS Div ($\downarrow$) | Inter. ($\uparrow$) | JS Div ($\downarrow$) | Inter. ($\uparrow$) | JS Div ($\downarrow$) | Inter. ($\uparrow$) |
| 0.15 | 0.2127 | 0.8087 | 0.1801 | 0.8359 | 0.1882 | 0.8145 | 0.1364 | 0.8590 | 0.1794 | 0.8295 |
| 0.30 | 0.1805 | 0.8517 | 0.1879 | 0.8303 | **0.1740** | **0.8292** | **0.1111** | **0.8893** | 0.1634 | 0.8501 |
| 0.45 | **0.1393** | **0.8941** | **0.1774** | **0.8414** | 0.1816 | 0.8211 | 0.1135 | 0.8826 | **0.1530** | **0.8598** |

# F  EFFICIENCY ANALYSIS OF SEQUENCE COMPRESSION AND ASYMMETRIC ARCHITECTURE

To investigate how our efficiency-related components influence the overall training efficiency, we conduct a detailed analysis of note-level sequence compression and the asymmetric architecture. Each component individually reduces computational cost, but their combination leads to a substantially larger improvement than using either one alone.

Table 8: Synergistic Efficiency Analysis of Sequence Compression and the Asymmetric Architecture. We report relative metrics where the baseline (6-6, Uncompressed) is set to 1.00x. Lower is better for VRAM, higher is better for Speed.

| Representation | Metric | Architecture | |
|---|---|---|---|
| | | 6-6 | 10-2 |
| Uncompressed | Training VRAM | 1.00x | 0.89x |
| | Training Speed | 1.00x | 1.07x |
| Compressed | Training VRAM | 0.63x | **0.38x** |
| | Training Speed | 1.81x | **3.13x** |

As shown in table 8, compression accelerates training by 1.81× on the 6–6 architecture, and the 10–2 architecture improves speed by 1.07× without compression. However, when both components are applied together, the overall training speed reaches 3.13×, substantially exceeding the product of their individual gains. A similar synergistic effect appears in VRAM reduction, where compression and architectural asymmetry jointly amplify memory savings.

These findings confirm that the proposed efficiency components are not merely additive but interact in a way that significantly amplifies their benefits—effectively achieving a "$1 + 1 > 2$" efficiency outcome.

# G  DETAILED DISCUSSION ABOUT ARCHITECTURAL TRADE-OFF.

We chose an asymmetric 10-2 architecture to balance performance and efficiency. To validate this choice, we compare it against a symmetric 6-6 baseline with a similar parameter count.

Table 9 shows the final rendering performance of both models after fine-tuning. While the symmetric 6-6 model achieves a slightly lower loss during pre-training, this does not translate to superior performance on the downstream rendering task. Our 10-2 model performs comparably to the 6-6 variant.

While the final rendering quality is highly comparable between the two architectures, the choice is justified by the significant gains in computational efficiency, detailed in Table 10. Our 10-2 model is over 2x faster in CPU inference and substantially more resource-efficient during training, requiring 40% less VRAM and achieving a 60% faster throughput.

Table 9: Objective evaluation results on the ASAP test set, comparing the asymmetric 10-2 architecture with a symmetric 6-6 baseline. Despite the 6-6 model having a deeper decoder, our 10-2 model achieves comparable or even slightly better performance on the final rendering task.

| Model | Velocity | | Duration | | IOI | | Pedal | | Overall (Avg.) | |
|---|---|---|---|---|---|---|---|---|---|---|
| | JS Div (↓) | Inter. (↑) | JS Div (↓) | Inter. (↑) | JS Div (↓) | Inter. (↑) | JS Div (↓) | Inter. (↑) | JS Div (↓) | Inter. (↑) |
| 6-6 | **0.1797** | 0.8411 | 0.2070 | **0.8427** | 0.1951 | 0.8116 | 0.1158 | 0.8837 | 0.1744 | 0.8448 |
| 10-2 | 0.1805 | **0.8517** | **0.1879** | 0.8303 | **0.1740** | **0.8292** | **0.1111** | **0.8893** | **0.1634** | **0.8501** |

Table 10: Efficiency comparison between the asymmetric (10-2) and symmetric (6-6) architectures. The 6-6 model is set as the baseline for relative speed and memory usage. Our design significantly accelerates both training and inference while reducing memory footprint.

| Metric | 10-2 | 6-6 | Advantage |
|---|---|---|---|
| CPU Inference Speed | 2.1x | 1.0x | 110% faster |
| Training VRAM | 0.6x | 1.0x | 40% less memory |
| Training Speed | 1.7x | 1.0x | 70% faster |

In conclusion, our asymmetric 10-2 architecture provides a superior trade-off, delivering state-of-the-art rendering quality while being significantly more efficient for both training and deployment. This makes it a more practical solution.

## H INFERENCE STRATEGY

During inference, our goal is to generate expressive performances that remain strictly matched to the input score. To preserve the note-level correspondence, we apply a hard pitch constraint: the model is free to sample expressive attributes but whenever a Pitch token is expected, we directly set it to the pitch from the input score instead of sampling. This enforces a one-to-one mapping between score notes and generated performance notes.

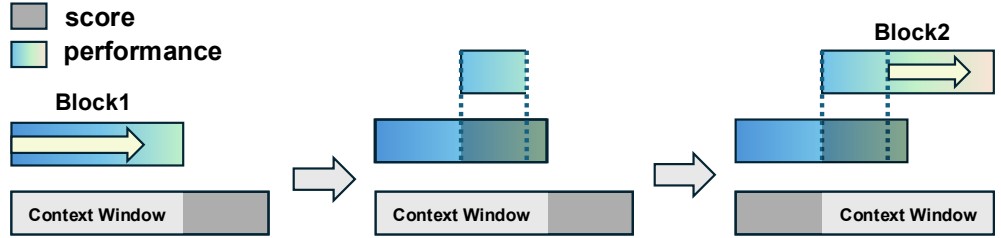

Figure 11: **Overlapped Block-wise Generation Strategy.** This figure illustrates how we generate long sequences using overlapping blocks. Block 1 is produced first. For the next block, we shift the window forward and reuse the stable overlapping region from the previous block as the decoder's context. Block 2 then continues generation from this context.

For pieces longer than 4096 tokens, we use an overlapped block-wise generation strategy illustrated in Figure 11. We first generate a 4096-token block. For the next block, we shift the window forward by 2048 tokens so that the new encoder input overlaps with the second half of the previous block. As decoder context, we reuse this overlapping region but drop a few unstable tokens at the tail before using it as the prompt. The model then continues generation from this context, and we append only the newly produced part. This procedure repeats until the piece is complete.

## I LIMITATIONS AND FUTURE WORK

While Pianist Transformer demonstrates strong performance, it has several limitations that suggest future directions. First, our efficient lightweight decoder is a performance bottleneck for scaling,

motivating research into more powerful yet efficient decoder architectures. Second, our focus on solo piano invites extending our self-supervised paradigm to multi-instrument and orchestral settings. Finally, moving beyond the limitation of score-based rendering to controllable generation from intuitive inputs like natural language remains a promising frontier.

