# OpenReview forum: "Pianist Transformer: Towards Expressive Piano Performance Rendering via Scalable Self-Supervised Pre-Training"
_ICLR.cc/2026/Conference — Submitted to ICLR 2026_

### Official Review · Reviewer_UJkA · 2025-10-26

**Soundness:** 2
**Presentation:** 3
**Contribution:** 3
**Rating:** 4
**Confidence:** 3

**Summary:**

This paper studies the problem of expressive piano performance rendering, which aims to generate a piano performance with expressive features (note velocity, dynamic timing, and pedal) given the score-only information. The proposed model features an encoder-decoder Transformer architecture and is trained across two-stage. It is first pre-trained on large-scale, unaligned performance-only data based on a mask language modelling objective. It is then fine-tuned on paired score-performance data in a seq2seq fashion. This strategy addresses the problem of scarce paired data, which has long been found in this field, and ablation study demonstrates significant contribution by pre-training. The resulting model is named “Pianist Transformer,” which also demonstrates superior performance compared to existing baselines through objective and subjective evaluation.

**Strengths:**

This is a well-written paper, with problem and goal clearly depicted and methodology well illustrated. A few strengths of the paper can be summarized as follows:

* **Unified Representation**: To make use of unaligned but more abundant data, the design of unified score/performance representation is well motivated in the paper. In both cases, note durations are represented as relative (inter onset) temporal units, which facilitate pre-training with performance-only data.

* **Comprehensive Treatment**: The paper provides a thorough and complete treatment of the expressive performance rendering task. It not only describes the model design and training pipeline in detail, but also includes a practical post-processing step to ensure compatibility with standard DAW workflows. Technical details are well documented throughout.

* **Demonstration Quality**: The attached synthesized demos sound convincing and musically natural, effectively supporting the evaluation results.

**Weaknesses:**

Despite its merits, the reviewer would like to raise several points of concern, the clarification or improvement of which could further strengthen the paper.

* **Baseline Comparisons**: While the paper includes two baseline models, both are relatively outdated. Incorporating more recent systems, such as (Borovik & Viro, 2023), would make the comparison more convincing and strengthen the evidence for the claimed performance improvements.

* **Limited Case Study Scale**: In Sections 4.4.2 and 4.4.3, the subjective evaluation highlights individual case studies across different music styles. However, these analyses appear to be based on only one or two pieces per style (Baroque, Classical, Romantic), which may introduce sampling bias. A more robust analysis could involve evaluating a larger number of pieces per style (maybe with objective metrics to make it more practical) to ensure generalizability.

**Questions:**

* **Metric Computation (Table 1)**: To the reviewer’s understanding, each metric (JS Div. and Inter.) is computed by comparing the model’s outputs against human performances. Are these metrics calculated per piece and then averaged, or are they derived from the aggregated distributions of all test pieces? Additionally, how are the “Human” results obtained in the table? (by comparing different human performances or through self-comparison?)

* **Subjective Evaluation (Figure 3)**: Does P4 correspond to the Liszt piece included in the supplementary demos? Figure 3 shows that the model’s rating on P4 is fairly higher than the human performance, whereas in the Liszt demo, the reviewer finds the human rendition sufficiently better.

* **Pedal Representation (Line 673)**: The explanation of the pedal encoding is somewhat unclear. Why are there four pedal tokens? And how are the 128 pedal values distributed among the four tokens?

* **Sequence Length**: How long is the input/output token sequence during inference? Is the model capable of generating longer sequences beyond this limit, for example by autoregressive continuation or segment stitching?

---

> ### Author Response · Authors · 2025-11-23
>
> Thank you for your careful review and constructive feedback on our work. Your comments are very valuable for improving the timeliness and rigor of our paper. We have conducted additional experiments accordingly, and we believe our responses address all your concerns.
>
> ## All modifications are marked in blue in the rivised paper.
>
> # Weakness 1: Lack of comparison with recent systems, e.g., ScorePerformer (Borovik & Viro, 2023)
> ## TL;DR:
> We added new comparison experiments, and our model significantly outperforms ScorePerformer on all objective metrics. See in Table 1 in the revised paper.
> ## Response:
> To ensure that our work is benchmarked against the latest progress in the field, we followed your suggestion and conducted additional experiments comparing our model directly with ScorePerformer (Borovik & Viro, 2023) using objective metrics.
>
> Why we did not originally include this baseline: ScorePerformer focuses on fine-grained, multi-level control of performance style. It relies on a style vector generated from a reference performance and user-specified instructions. In contrast, our work focuses on generating a high-quality, human-like performance directly from the score, aiming to validate the effectiveness of large-scale self-supervised pretraining on this rendering task. The ScorePerformer paper mainly evaluates stylistic controllability and does not provide direct comparisons of rendering quality with existing performance-rendering models such as VirtuosoNet, which reflects the different focuses of prior work.
>
> Under the scenario where only the score is available and no reference performance is provided, the ScorePerformer paper suggests sampling the style vector from a prior distribution for unconditional generation. We follow this recommended procedure in our comparison experiments. The results are as follows:
>
> | **Model**                | **Velocity JS↓** | **Velocity Inter↑** | **Duration JS↓** | **Duration Inter↑** | **IOI JS↓** | **IOI Inter↑** | **Pedal JS↓** | **Pedal Inter↑** | **Overall JS↓** | **Overall Inter↑** |
> |--------------------------|------------------|---------------------|------------------|---------------------|-------------|----------------|----------------|-------------------|------------------|---------------------|
> | ScorePerformer           | 0.4004           | 0.6256              | 0.3116           | 0.7126              | 0.4555      | 0.6071         | 0.6174         | 0.4164            | 0.4462           | 0.5904              |
> | **Pianist Transformer**  | **0.1805**       | **0.8517**          | **0.1879**       | **0.8303**          | **0.1740**  | **0.8292**     | **0.1111**         | **0.8893**            | **0.1634**       | **0.8501**          |
>
> As shown, Pianist Transformer significantly outperforms ScorePerformer across all objective metrics. We believe this further highlights the central advantage of our work: through large-scale self-supervised pretraining, our model learns an intrinsic prior for expressive performance, enabling it to generate expressive results even without external stylistic guidance.
>
> We appreciate your suggestion. This comparison strengthens both the contribution and the timeliness of our work. We have added the results to the revised paper (see Table 1).
>
> # Weakness 2: Limited scale of subjective evaluation and potential sampling bias
> ## TL;DR:
> We acknowledge that the subjective evaluation is limited in scale, but its conclusions are highly consistent with the objective metrics computed over the entire test set, supporting the superiority of our model.
> ## Response:
> We fully understand your concern regarding generalizability and acknowledge that a larger-scale subjective evaluation would indeed provide stronger evidence.
>
> Our current design aims to balance feasibility and rigor. Subjective listening tests require recruiting many participants, and to avoid fatigue effects, each participant can only evaluate a limited number of samples. In our study, we selected six musical pieces spanning different historical periods and styles, and each participant evaluated five different versions (four versions for the out-of-domain piece, which lacks human MIDI performances). This is the largest scale feasible within our available resources.
>
> To compensate for the limited breadth of the subjective study, all objective metrics (Table 1) were computed over the entire ASAP test set. These objective results exhibit trends highly consistent with the subjective evaluation, jointly providing strong evidence for the superiority of our model.

---

> ### Author Response · Authors · 2025-11-23
>
> # Question 1: How are the objective metrics computed, and how are the "Human" results obtained?
> ## TL;DR:
> All objective metrics are computed from aggregated distributions. "Human" results are obtained by randomly selecting one human performance for each piece and comparing it with the remaining human performances. See in Appendix B.4 in the revised paper.
> ## Response:
> Our objective metrics are computed as follows:
> 1. Metric computation: For each expressive dimension, we aggregate all model-generated performances and all corresponding human performances across the entire test set to form two global distributions. We then compute the JS divergence and Intersection Area between these distributions.
> 2. Source of "Human" results: Each piece in the ASAP dataset contains multiple human performances. Our "Human" metric is computed using a leave-one-out strategy: for each piece, we randomly select one human performance as the generated sample, while the remaining human performances form the reference distribution. We compute the same metrics between these distributions, reflecting the natural variation among human performers.
>
> We have added this description to Appendix B.4 in the revised paper.
>
> # Question 2: Does P4 correspond to a Liszt composition in Figure 3?
> ## TL;DR:
> P4 in Figure 3 corresponds to a Bach piece. The Liszt piece was used for consistency analysis, as detailed in Appendix D.4.
> ## Response:
> P4 in Figure 3 corresponds to J. S. Bach, Prelude and Fugue in G minor, BWV 885 (Prelude), not the Liszt piece.
> The Liszt piece you mentioned (Étude d’exécution transcendante No. 1) was used for consistency analysis. As described in Appendix D.4, we inserted two identical copies of the audio to assess reliability of participant ratings. For this purpose, this piece does not appear in Figure 3 or other main results.
>
> We apologize for the earlier lack of clarity. We have labeled the composer for each piece in  Figure 4 in the revised paper.
>
> # Question 3: Why are there four pedal tokens, and how are the 128 pedal values distributed among them?
> ## TL;DR:
> Pedal is a continuous curve. We sample four evenly spaced points between two consecutive notes. See in Appendix B.2.2 in the revised paper.
> ## Response:
> We agree that the pedal encoding was not sufficiently clear. The pedal signal in piano performance is a continuous curve (values 0–127). To integrate it into our note-centric discrete token sequence, we adopt a sampling-based approach.
>
> For each note in the score, there is a time interval before the next note. We sample four evenly spaced time points within this interval and record the corresponding pedal values (0–127) as the tokens Pedal1, Pedal2, Pedal3, and Pedal4.
>
> We choose four samples because they capture essential pedaling techniques such as "after-pedaling" and "syncopated pedaling". We have added a concrete example in Appendix B.2.2 to help readers better understand our tokenization scheme in revised paper.
>
> # Question 4: What is the maximum token sequence length during inference, and can the model generate sequences exceeding this limit?
> ## TL;DR:
> The maximum input/output context length is 4096 tokens during inference. For longer pieces, we generate using overlapping sliding windows. See in Appendix H in the revised paper.
> ## Response:
> During both training and inference, our model’s maximum sequence length is 4096 tokens.
> For pieces requiring more than 4096 tokens, we use an overlapping blockwise autoregressive generation strategy:
> 1. Generate the first block (4096 tokens).
> 2. Prepare the next block:
>  - Encoder input: slide forward by 2048 tokens to form the next 4096-token sequence, whose first half overlaps with the previous block’s second half.
>  - Decoder context: use the second half of the previous generated block (excluding a small unstable portion at the end) as prefix tokens.
> 3. Generate and stitch: generate the non-overlapping second half of the new block and concatenate it with previous results.
> 4. repeat until the entire piece is generated.
>
> Using this approach, the model can generate complete musical pieces of arbitrary length while maintaining smooth transitions between blocks. All test samples were generated using this method. A detailed description and an illustrative figure have been added in Appendix H.
>
> Thank you again for your time and expert feedback. Your suggestions have helped us significantly improve the paper. We hope our responses address your concerns and are happy to discuss if you have any further questions!

---

### Official Review · Reviewer_D9R1 · 2025-10-30

**Soundness:** 2
**Presentation:** 3
**Contribution:** 2
**Rating:** 4
**Confidence:** 4

**Summary:**

This paper presents Pianist Transformer, a model for expressive piano performance rendering. The authors propose (1) leveraging unaligned performance MIDI for pretraining, (2) introducing a unified representation for both score and performance MIDI, and (3) designing an efficient asymmetric Transformer architecture. Both objective metrics and subjective listening evaluations demonstrate the benefits of the proposed pretraining stage.

**Strengths:**

1. The incorporation of self-supervised learning on unpaired performance MIDI data is meaningful and could be extended to other symbolic music tasks beyond piano rendering.
2. The experimental section is thorough, particularly the design and analysis of the subjective evaluation.

**Weaknesses:**

1. Two of the major claimed contributions overlap with prior work. Existing tokenization schemes such as REMI [1], MIDI-Like [2], CPWord [3], and Octuple [4] already represent pitch, duration, and velocity as discrete tokens. Moreover, the proposed note-level compression in the encoder resembles CPWord and Octuple designs, where each note is represented by a compression of fixed number of tokens (e.g., MusicBERT [4]). The authors should clarify how their unified representation provides new capabilities beyond these established methods.
2. There is insufficient discussion on the efficiency-performance trade-off. In Section 4.5, the symmetric 6-6 architecture achieves lower pretraining validation loss than the proposed asymmetric 10-2 model, suggesting that efficiency gains may come at the cost of performance. However, there is no detailed comparison of their rendering performance (objective or subjective) or inference efficiency. Considering that the model is relatively small (~0.1B parameters), it is unclear whether efficiency should be a primary concern, and whether potential performance degradation, if any, is justified by the speedup.
3. As one of the core components of the proposed framework, the pretraining setup lacks sufficient detail regarding the masking strategy; specifically, the mask rate, the masking pattern (e.g., random tokens vs. consecutive notes), and how these choices influence learning. An ablation study examining the impact of different mask rates on overall performance would strengthen the paper.

References

[1] Huang and Yang, “Pop Music Transformer: Beat-based Modeling and Generation of Expressive Pop Piano Compositions,” ACM MM, 2020.

[2] Oore et al., “This Time with Feeling: Learning Expressive Musical Performance,” Neural Computing and Applications, 2020.

[3] Hsiao et al., “Compound Word Transformer: Learning to Compose Full-Song Music over Dynamic Directed Hypergraphs,” AAAI, 2021.

[4] Zeng et al., “MusicBERT: Symbolic Music Understanding with Large-Scale Pre-Training,” ACL-IJCNLP Findings, 2021.

**Questions:**

Could the authors please addressing the following:

1. What are the fundamental differences between the proposed unified MIDI representation and prior tokenizations (e.g., REMI, Octuple, CPWord)?
2. The PDMX dataset contains many multi-instrument scores rather than solo piano parts. How are these multi-track scores handled during pretraining?
3. During pretraining, data from both score-based (e.g., PDMX) and performance-based (e.g., POP909) MIDI are used. Are they explicitly distinguished in the model input? If not, how does the model learn to differentiate the reconstruction objective during pretraining from the rendering objective during inference when a score MIDI is provided?
4. In Figure 3 (b), why does VirtuosoNet-ISGN perform worse than the “Score” baseline in most of the cases, despite being a strong model in previous literature?
5. In Algorithm 1, the method requires aligned note pairs from score and performance sequences. How is this alignment established, especially under one-to-many senarios such as ornamentations?

---

> ### Author Response · Authors · 2025-11-23
>
> Thank you for your careful review and constructive comments on our work. Your professional feedback has prompted us to further refine the core contributions and technical details of our method. We have conducted targeted supplementary experiments, and we believe our responses address all of your concerns.
>
> ## All modifications are marked in blue in the rivised paper.
>
> # Weakness 1 / Question 1: Two main contributions overlap with prior work. The proposed representation should more clearly articulate its novelty and advantages over existing methods.
> ## TL;DR:
> The core innovation of our representation is that it does not rely on score-level musical structures, enabling large-scale pretraining on massive unaligned performance datasets while remaining note-centric and well-suited to performance rendering. The note-centric compression is not an isolated innovation. When combined with the proposed 10-2 asymmetric architecture, it leads to substantial computational savings (62% memory reduction and a 3.13× training speedup). See in Appendix B.2.3 and Appendix F in the revised paper.
> ## Response:
> Our proposed representation differs fundamentally from REMI, CPWord, Octuple, and similar approaches in that it does not require any high-level musical structure (such as bars, beats, or tempo). Instead, we encode temporal information directly using inter-onset intervals (IOI) between notes. In contrast:
> - REMI, CPWord, and Octuple represent timing using Bar and Position, which require bar-level information computed from the MIDI file and strict quantization of notes—conditions typically satisfied only by score MIDIs.
> - MIDI-Like treats MIDI as a stream of events rather than a note-centric structure, making it poorly suited for performance rendering where precise note alignment is essential.
>
> Our note-centric IOI representation brings decisive advantages:
>
> 1. Large-scale pretraining: Performance MIDI lacks bars, beats, and tempo, so formats like REMI can’t handle it or unify it with score MIDI. Our IOI-based design supports both, enabling large-scale self-supervised pretraining that greatly strengthens score-MIDI representations.
> 2. Note-centric alignment and efficiency optimization: Our representation is explicitly note-centered, allowing precise note-level alignment for supervised fine-tuning and enabling strict score-following by constraining pitches during inference. This note-level structure also provides significant compression and efficiency gains.
>
> Additionally, REMI, MIDI-Like, CPWord, and Octuple do not encode pedal information, resulting in generated piano performances that lack the resonance effects essential for realistic piano sound.
>
> Our note-centric IOI representation brings decisive advantages:
>
> | **Representation** | **Temporal Representation** | **Note-centric** | **Encodes Pedal** |
> |--------------------|-----------------------------|-------------------|--------------------|
> | REMI               | Bar & Pos                   | No                | No                 |
> | MIDI-Like          | Time shift                  | No                | No                 |
> | CPWord             | Bar & Pos                   | No                | No                 |
> | Octuple            | Bar & Pos                   | Yes               | No                 |
> | **Ours**           | **Time shift**              | **Yes**           | **Yes**            |
>
> Regarding the note-level compression on the encoder side, we acknowledge that its form bears some resemblance to CPWord and similar methods. However, it is not an isolated innovation. It is introduced jointly with our 10-2 asymmetric architecture as a pair of synergistic efficiency-oriented components. When used together, these two components lead to an efficiency gain that is greater than the sum of their individual improvements, as illustrated in the table below:
> | **Representation** | **Metric**        | **6-6** | **10-2** |
> |--------------------|-------------------|--------:|---------:|
> | **Uncompressed**   | Training VRAM     | 1.00x   | 0.89x    |
> |                    | Training Speed    | 1.00x   | 1.07x    |
> | **Compressed**     | Training VRAM     | 0.63x   | **0.38x** |
> |                    | Training Speed    | 1.81x   | **3.13x** |
>
> The fact that their combined efficiency surpasses the additive effect of using each component independently demonstrates that they work synergistically rather than independently. This constitutes an innovation at the system level.
>
> We have added a dedicated subsection in the appendix of the revised paper to compare different representation schemes and analyze this synergistic effect in detail. Please refer to Appendix B.2.3 and Appendix F.

---

> ### Author Response · Authors · 2025-11-23
>
> # Weakness 2: Insufficient discussion on performance–efficiency trade-offs
> ## TL;DR:
> We conducted additional experiments. Compared with the symmetric 6-6 model, our asymmetric 10-2 model shows almost no loss in rendering quality, while delivering 110% faster CPU inference, 40% lower training memory consumption, and 70% faster training speed. See in Sections 4.5 and Appendix G in the revised paper.
> ## Response:
> To thoroughly evaluate the relationship between efficiency and performance, we followed your suggestion and conducted an additional experiment directly comparing our asymmetric 10-2 model with a symmetric 6-6 model on the final rendering task using objective metrics. As shown in the table below, although the 6-6 model achieves slightly lower validation loss during pretraining, the two models exhibit very similar objective performance on the final rendering task. In fact, the 10-2 model is marginally better overall.
> | **Model** | **Velocity JS↓** | **Velocity Inter↑** | **Duration JS↓** | **Duration Inter↑** | **IOI JS↓** | **IOI Inter↑** | **Pedal JS↓** | **Pedal Inter↑** | **Overall JS↓** | **Overall Inter↑** |
> |-----------|------------------|---------------------|------------------|---------------------|-------------|----------------|----------------|-------------------|------------------|---------------------|
> | **6-6**   | **0.1797**       | 0.8411              | 0.2070           | **0.8427**          | 0.1951      | 0.8116         | 0.1158         | 0.8837            | 0.1744           | 0.8448              |
> | **10-2**  | 0.1805           | **0.8517**          | **0.1879**       | 0.8303              | **0.1740**  | **0.8292**     | **0.1111**     | **0.8893**        | **0.1634**       | **0.8501**          |
>
> This indicates that the small advantage gained by a deeper decoder during pretraining does not translate into meaningful improvements in final rendering quality. Our 10-2 architecture is already sufficient for producing high-quality performance renderings. It is possible that the benefit of a deeper decoder would require substantially more training before it could meaningfully affect downstream rendering results.
>
> One of our primary goals is to design a model that is not only effective but efficient for practical deployment, including on resource-limited devices (e.g., CPU-only environments). In terms of efficiency, the 10-2 architecture provides clear benefits:
>
> | **Metric**               | **10-2** | **6-6** | **Advantage**        |
> |--------------------------|---------:|--------:|-----------------------|
> | CPU Inference Speed      | 2.1x     | 1.0x    | 110% faster           |
> | Training VRAM            | 0.6x     | 1.0x    | 40% less memory       |
> | Training Speed           | 1.7x     | 1.0x    | 70% faster            |
>
> Overall, our asymmetric 10-2 architecture achieves a highly desirable performance–efficiency balance. It preserves nearly all rendering quality while offering more than 2× inference speedup and significantly reducing training resource requirements. We believe that for real-world applications, such substantial efficiency gains are more valuable than the minor differences in performance.
>
> We have incorporated these experimental results and discussions into Sections 4.5 and Appendix G of the revised paper.

---

> ### Author Response · Authors · 2025-11-23
>
> # Weakness 3: Insufficient disclosure of pretraining details and mask-rate ablation is needed.
> ## TL;DR:
> We adopted standard practice to validate the paradigm itself using a masking ratio of 0.3 with a random token masking strategy. Following your suggestion, we performed an ablation study, which shows that the pretraining approach consistently outperforms pure supervised learning across all mask rates. See in Appendix B.3.1 and Appendix E in the revised paper.
> ## Response:
> Our pretraining setup uses a masking ratio of 0.3 with a random token masking strategy. This choice follows established practices in recent NLP literature [1], aiming for a robust and widely validated baseline configuration.
>
> Our overarching goal is to build a practical and scalable framework for piano performance rendering. This goal manifests at two levels:
> 1. Algorithmic perspective, validating the paradigm itself: The central objective of our work is to verify whether large-scale self-supervised pretraining can bring fundamental performance improvements to this domain. Therefore, we intentionally choose a simple and standard masking objective, rather than adopting sophisticated masking strategies that may only provide gains for narrow tasks. This ensures that the model’s strong performance can be attributed more clearly to the power of data scale and the pretraining paradigm itself, rather than to specialized training tricks.
> 2. Engineering perspective, designing an efficient architecture for real-world use: Having validated the paradigm’s effectiveness, we also care about practical usability. As discussed in our response to Weakness 2, our carefully designed 10-2 asymmetric architecture aims to significantly improve training and inference efficiency without sacrificing rendering quality, enabling deployment across a wider range of scenarios.
>
> Under this classical baseline setting, our method already demonstrates substantial improvements over pure supervised learning. We agree that exploring how specific pretraining strategies affect performance is meaningful. Following your recommendation, we conducted an ablation study on different masking ratios, with results shown below:
>
> | **Mask Ratio** | **Velocity JS↓** | **Velocity Inter↑** | **Duration JS↓** | **Duration Inter↑** | **IOI JS↓** | **IOI Inter↑** | **Pedal JS↓** | **Pedal Inter↑** | **Overall JS↓** | **Overall Inter↑** |
> |----------------|------------------|---------------------|------------------|---------------------|-------------|----------------|----------------|-------------------|------------------|---------------------|
> | 0.15           | 0.2127           | 0.8087              | 0.1801           | 0.8359              | 0.1882      | 0.8145         | 0.1364         | 0.8590            | 0.1794           | 0.8295              |
> | 0.30           | 0.1805           | 0.8517              | 0.1879           | 0.8303              | **0.1740**  | **0.8292**     | **0.1111**     | **0.8893**        | 0.1634           | 0.8501              |
> | 0.45           | **0.1393**       | **0.8941**          | **0.1774**       | **0.8414**          | 0.1816      | 0.8211         | 0.1135         | 0.8826            | **0.1530**       | **0.8598**          |
>
> The experimental results show that our chosen 30% masking ratio significantly outperforms 15% and is slightly below 45%. This suggests that classical masking practices from NLP may not transfer directly to piano performance rendering; the nature of different tasks may influence optimal pretraining strategies. This is an interesting direction for future work. Nevertheless, under all settings, introducing pretraining yields large gains over pure supervised learning. This supports our core claim: large-scale self-supervised pretraining can bring fundamental improvements to piano performance rendering without relying on complex masking strategies.
>
> We have included the ablation results and the corresponding discussion in Appendix B.3.1 and Appendix E of the revised paper.
>
> [1] Benjamin Warner, Antoine Chaffin, Benjamin Clavié, Orion Weller, Oskar Hallström, Said Taghadouini, Alexis Gallagher, Raja Biswas, Faisal Ladhak, Tom Aarsen, Griffin Thomas Adams, Jeremy Howard, Iacopo Poli: Smarter, Better, Faster, Longer: A Modern Bidirectional Encoder for Fast, Memory Efficient, and Long Context Finetuning and Inference. ACL (1) 2025: 2526-2547

---

> ### Author Response · Authors · 2025-11-23
>
> # Question 1: What are the fundamental differences between our representation and prior tokenizations?
> Please refer to our response to Weakness 1.
> # Question 2: How do we handle multi-track score MIDI during pretraining?
> ## TL;DR:
> We standardize all MIDI: merging tracks, removing duplicated notes, and normalizing tempo, so the model can learn general musical grammar. See in Appendix B.1 in the revised paper.
> ## Response:
> During pretraining, to allow the model to learn from as broad a musical corpus as possible, we apply a unified normalization pipeline to all MIDI data (including multi-track scores such as PDMX). The preprocessing steps are:
> 1. Track merging: We merge all tracks in scores into a single monophonic note sequence.
> 2. Duplicate removal: To avoid introducing noise, we remove duplicate notes that share the same pitch and onset time after track merging.
> 3. Normalization: We set the tempo of all MIDI files to a unified value (120 BPM) and proportionally normalize note durations and timing.
>
> Our goal in the pretraining stage is to enable the model to learn general musical patterns. Thus, we treat multi-instrument scores as a rich source of musical material, and the normalization above allows them to be incorporated consistently into our unified pretraining framework.
>
> We have added these preprocessing details to the revised paper. Please refer to Appendix B.1.
> # Question 3: Do we differentiate between score MIDI and performance MIDI during pretraining? If not, how does the model distinguish pretraining vs. downstream objectives?
> ## TL;DR:
> We do not distinguish them during pretraining. Pretraining teaches musical representation; the score-to-performance mapping is learned only during supervised fine-tuning. See in Appendix B.1 in the revised paper.
> ## Response:
> During pretraining, we do not explicitly distinguish score MIDI from performance MIDI. The sole objective of this stage is for the model to learn the grammar and structural patterns of music from large-scale data. Specifically, the encoder learns to build meaningful musical representations, and the decoder learns to generate coherent musical sequences. At this point, the model does not learn score-to-performance mapping.
>
> This mapping capability is learned entirely during the SFT stage. In the SFT stage, we train only on high-quality score–performance aligned pairs. Through this process, the model learns to translate the symbolic score representation into expressive human-like performance attributes.
>
> As shown in Fig. 3(c) of the revised paper, a well-pretrained model quickly acquires this translation ability, with losses converging rapidly during fine-tuning. This strongly validates the effectiveness of our two-stage strategy.

---

> ### Author Response · Authors · 2025-11-23
>
> # Question 4: Why does ISGN perform worse than Score in many cases?
> ## TL;DR:
> ISGN exhibits strong style dependence. It performs well on Romantic and modern pop pieces but poorly on Classical and Baroque pieces, sometimes worse than an unrendered Score. See in  Figure 4 in the revised paper.
> ## Response:
> We analyzed this phenomenon and found that VirtuosoNet-ISGN’s performance is highly style-dependent.
> The subjective evaluation covers multiple musical periods: Baroque, Classical, Romantic, and modern pop (as shown in Figure 3(b) and Figure 8). Specifically, P1 and P2 are Classical pieces, P3 is Romantic, P4 is Baroque, and Figure 8 is modern pop. Our observations:
> - Strong performance: On Romantic and modern pop pieces, VirtuosoNet-ISGN performs reasonably well and is sometimes competitive. These styles typically embrace freer tempo rubato and more extensive pedal usage.
> - Weak performance: In Classical and Baroque pieces, ISGN often underperforms, even compared to the unrendered Score baseline. These styles favor structural clarity, rhythmic stability, and controlled dynamics. Since Score has perfectly stable timing and no expressive deviations, its neutral version inadvertently fits the stylistic expectations better. In contrast, ISGN may introduce stylistically inappropriate rubato or dynamics. In these genres, incorrect expression can be worse than no expression, causing its ratings to fall below the Score baseline.
>
> Thus, ISGN’s overall ranking is dragged down by its poor performance on certain styles. In contrast, our model, similar to human performers, maintains consistently high quality across all styles, demonstrating stronger robustness and generalization ability. We mark the composer for each excerpt in Figure 4 of the revised paper.
>
> # Question 5: Algorithm 1 requires alignment between the performance and score sequences. How is this achieved, especially in one-to-many scenarios?
> ## TL;DR:
> Algorithm 1 operates on the model-generated performance and the input score. During inference, pitch is constrained to enforce strict one-to-one alignment between the generated performance and the input score. See in  Appendix H in the revised paper.
> ## Response:
> The primary purpose of Algorithm 1 is to convert the model-generated performance, represented in absolute time, into a MIDI file that is strictly aligned to the original score, with tempo changes encoded via a tempo track, enabling convenient editing in DAWs.
>
> During inference, to ensure that the rendering stays faithful to the score, the decoder is forced to use the pitch of the corresponding score note at each note. The model is only allowed to predict expressive attributes such as velocity, duration, IOI, and pedal usage.
> This guarantees a strict one-to-one note-level alignment between the output performance and the input score. As a result, the model generates an interpretation of the given score rather than a composition with additional notes. This design ensures both usability and fidelity.
>
> We have added detailed explanations of this mechanism and inference strategy in the revised paper, see Appendix H.
> Thank you again for your time and expert feedback. Your suggestions have helped us significantly improve the paper. We hope our responses address your concerns and are happy to discuss if you have any further questions!

---

### Official Review · Reviewer_HbNE · 2025-11-01

**Soundness:** 4
**Presentation:** 3
**Contribution:** 3
**Rating:** 6
**Confidence:** 4

**Summary:**

This paper presents a novel piano performance rendering model together with a training strategy and a unified tokenization that handles both score and performance MIDI. Using this tokenization, the model is first pretrained with a masked language modeling objective that predicts masked attributes from either score or performance MIDI. It is then supervisedly fine-tuned to render performance from score. The experiments report superior performance over baselines.

**Strengths:**

1. The tokenization and the overall pipeline are very well-designed and make sense in how they help with performance rendering tasks.
2. The experiment design is clear and evaluates the model from multiple perspectives.
3. The provided listening samples are very convincing, showing massive improvement compared to existing methods.

**Weaknesses:**

1. Opening with an experiment figure feels a bit off and does little to support the narrative. The results in Figure 1, especially the variant without pretraining, should be detailed in the experiments section with fuller analysis.
2. The paper could more intuitively explain token representations and model I/O at each stage. Small, concrete examples would improve readability.
3. The paper’s relevance to the ICLR community is under-articulated; as written, it reads more naturally for a computer-music venue. Section 4.5’s discussion feels loosely connected to the core contributions. A more urgent question to ask is perhaps: by doing pre-training, which performance criteria benefit, compared to the one without pre-training. Also, would large-scale pretraining under the SSL objective be superior to supervised learning on large-scale transcribed/quantized audio?

**Questions:**

See weaknesses.

---

> ### Author Response · Authors · 2025-11-23
>
> Thank you for your positive assessment of our work and for the constructive and insightful comments. You highlighted several key points that can further improve the clarity and impact of our paper. Following your suggestions, we have revised the paper accordingly, and we believe these changes significantly enhance the overall quality.
> ## All modifications are marked in blue in the rivised paper.
> # Weakness 1: The experimental figure should be moved to the experimental section and explained in more detail.
> ## TL;DR:
> We have moved Figure 1 to the experimental section and placed a comparison figure of the supervised paradigm vs. the self-supervised pretraining and fine-tuning paradigm in the introduction. See in Introduction and Section 4.2 in the revised paper.
> ## Response:
> We fully agree. Our original intention for placing the key experimental result at the beginning was to immediately convey the substantial benefits of large-scale pretraining and to motivate readers about our central argument.
>
> After considering your suggestion, we realized that the previous presentation may have appeared abrupt. In the revised version, we replaced the original Figure 1 in the introduction with a new conceptual comparison diagram illustrating the supervised paradigm vs. the self-supervised pretraining and fine-tuning paradigm, thereby motivating our central idea more naturally.
>
> We then moved the original Figure 1 into the experimental section and expanded the analysis comparing the pretrained and non-pretrained models in Section 4.2. Please refer to the updated Introduction and Section 4.2.
> # Weakness 2: The explanation of token representation and the model’s I/O at each stage should be more intuitive.
> ## TL;DR:
> We added a concrete example of token representation and enriched the description of Figure 2 with more intuitive explanations of the model’s I/O at each stage. See in Appendix B.2.2 and Figure 2 in the revised paper.
> ## Response:
> This is a good suggestion. We acknowledge that our previous explanation of data representation was overly technical and lacked a concrete example to help readers intuitively understand the method.
>
> In the revised version, we added a new subsection in the appendix. In this subsection, we use a short, simple musical score as a concrete example to demonstrate how it is converted into a token sequence. We also updated Figure 2 to explicitly illustrate the model’s inputs and outputs (or targets) at each stage:
> * Pretraining: The model receives a masked or corrupted token sequence and is trained to recover the original sequence.
> * SFT stage: The model takes a score-token sequence as input and learns to predict the corresponding performance-token sequence.
> * Inference: The model is given score tokens and generates performance tokens as output.
>
> We believe this example and the clarifications improve the paper’s readability and make our core mechanism clearer. Please refer to Appendix B.2.2 and Figure 2 in the updated paper.

---

> ### Author Response · Authors · 2025-11-23
>
> # Weakness 3: The paper does not clearly articulate its relevance to the ICLR community.
> ## TL;DR:
> Our core contribution is the first successful application of the self-supervised pretraining and fine-tuning paradigm to piano performance rendering, a complex structured-generation task, representing a paradigm shift in this field. This aligns closely with ICLR’s focus on self-supervised learning and representation learning. Moreover, AI-for-Music has an active audience at ICLR and other general AI community.
> ## Response:
> We agree that clarifying our contribution to the ICLR community is essential. Our core contribution is a paradigm transfer: we successfully apply the widely validated large-scale self-supervised pretraining and fine-tuning paradigm from NLP/CV to piano performance rendering, which is a complex, non-linguistic, fine-grained expressive conditional-generation problem.
> Following your suggestion, we added a conceptual comparison figure in the introduction to more explicitly position our work as an instance of this general learning-paradigm transfer. We believe this aligns strongly with ICLR’s central themes in self-supervised learning and representation learning. Furthermore, Section 4.5 analyzes whether scaling-law-like phenomena observed in NLP also appear in piano-performance rendering.
> We also note that AI-for-Music research has been well-represented at ICLR (e.g., [1], [2], [3]), including an Oral and a Spotlight paper, and work on piano performance rendering has appeared as an Oral at ICML [4]. NeurIPS 2025 is also hosting an AI-for-Music workshop. Thus, we believe our insights fit naturally within the ICLR community.
> In addition, regarding your two specific questions:
> 1. Benefits brought by pretraining: Pretraining yields comprehensive improvements. As shown in Table 1 and Figure 3 in the updated version, the pretrained model significantly outperforms the one without pretraining on objective metrics (overall JS divergence reduced by 60.2%), starts SFT with lower initial loss, converges faster, and reaches a better final loss.
> 2. Self-supervised pretraining vs. large-scale supervised data via transcription/quantization: While creating large supervised datasets via transcription or quantization is theoretically possible, in practice this requires a reliable model or tool to convert expressive performance MIDI into score MIDI. Simple note-quantization tools cannot adequately extract the rhythmic backbone from free expressive performances. In the early stage, we experimented with training such a quantization model using the pretrained model in reverse on SFT data, but the results were unsatisfactory. We believe this is an interesting open question. Therefore, for present work, we chose the more direct path of self-supervised learning.
>
> Thank you again for your time and expert feedback. Your suggestions have helped us significantly improve the paper. We hope our responses address your concerns and are happy to discuss if you have any further questions!
>
> [1] Curtis Hawthorne, Andriy Stasyuk, Adam Roberts, Ian Simon, Cheng-Zhi Anna Huang, Sander Dieleman, Erich Elsen, Jesse H. Engel, Douglas Eck: Enabling Factorized Piano Music Modeling and Generation with the MAESTRO Dataset. ICLR 2019.
>
> [2] Ziyu Wang, Lejun Min, Gus Xia: Whole-Song Hierarchical Generation of Symbolic Music Using Cascaded Diffusion Models. ICLR 2024
>
> [3] Louis Bradshaw, Simon Colton: Aria-MIDI: A Dataset of Piano MIDI Files for Symbolic Music Modeling. ICLR 2025.
>
> [4] Dasaem Jeong, Taegyun Kwon, Yoojin Kim, Juhan Nam: Graph neural network for music score data and modeling expressive piano performance. ICML 2019.

---

### Meta-Review · Area_Chair_7PSg · 2026-01-06

**Summary:**

This paper proposes Pianist Transformer for expressive piano performance rendering, using a unified score/performance MIDI representation and a two-stage pipeline: masked self-supervised pre-training on large-scale unpaired MIDI followed by supervised score-to-performance fine-tuning. Reviewers found the overall approach well-motivated and the demos/results compelling, especially the use of abundant unpaired performance MIDI to mitigate limited paired supervision.

**Reviewer Concerns:**

Key concerns in the initial reviews centered on (i) novelty/overlap with prior tokenizations and note-level compression designs, (ii) the efficiency–quality trade-off of the asymmetric 10-2 architecture, and (iii) evaluation scope (outdated baselines and limited per-style subjective case studies).

**Reviewer Scores:**

This paper initially received mixed reviews, with 2x marginally below the acceptance threshold and 1x marginally above the acceptance threshold. In rebuttal, the authors clarified that their representation does not rely on bars/beats/tempo and instead uses IOI-based timing, and they described explicit pedal encoding. They also added new experiments: a direct 10-2 vs 6-6 downstream comparison showing very similar objective rendering quality alongside substantial efficiency gains; a mask-rate ablation study; and an additional recent baseline (ScorePerformer) on objective metrics, where Pianist Transformer outperforms it. While these additions strengthen the paper and address parts of (ii) and (iii), the remaining sticking points for this decision are whether the representation/architecture provides sufficiently clear methodological novelty beyond enabling scalable SSL pretraining, and whether the “human-level” claim is fully supported given that the subjective study is relatively limited in breadth even if the authors argue consistency with objective metrics over the full test set.
Considering the above, I recommend rejection.

---

### Decision · Program_Chairs · 2026-01-26

Reject